# Acetyl-CoA flux regulates the proteome and acetyl-proteome to maintain intracellular metabolic crosstalk

Inca A. Dieterich[1,2,3], Alexis J. Lawton [4,13], Yajing Peng[1,2,13], Qing Yu[5,10,13], Timothy W. Rhoads [1], Katherine A. Overmyer [6,7], Yusi Cui[5], Eric A. Armstrong[4], Porsha R. Howell[1,11], Maggie S. Burhans[1,12], Lingjun Li [5], John M. Denu[4], Joshua J. Coon[6,7], Rozalyn M. Anderson[1,8] & Luigi Puglielli [1,2,8,9]

AT-1/SLC33A1 is a key member of the endoplasmic reticulum (ER) acetylation machinery, transporting acetyl-CoA from the cytosol into the ER lumen where acetyl-CoA serves as the acetyl-group donor for Nε-lysine acetylation. Dysfunctional ER acetylation, as caused by heterozygous or homozygous mutations as well as gene duplication events of *AT-1/SLC33A1*, has been linked to both developmental and degenerative diseases. Here, we investigate two models of AT-1 dysregulation and altered acetyl-CoA flux: AT-1$^{S113R/+}$ mice, a model of AT-1 haploinsufficiency, and AT-1 sTg mice, a model of AT-1 overexpression. The animals display distinct metabolic adaptation across intracellular compartments, including reprogramming of lipid metabolism and mitochondria bioenergetics. Mechanistically, the perturbations to AT-1-dependent acetyl-CoA flux result in global and specific changes in both the proteome and the acetyl-proteome (protein acetylation). Collectively, our results suggest that AT-1 acts as an important metabolic regulator that maintains acetyl-CoA homeostasis by promoting functional crosstalk between different intracellular organelles.

[1] Department of Medicine, University of Wisconsin-Madison, Madison, WI, USA. [2] Waisman Center, University of Wisconsin-Madison, Madison, WI, USA. [3] Neuroscience Training Program, University of Wisconsin-Madison, Madison, WI, USA. [4] Department of Biomolecular Chemistry and the Wisconsin Institute for Discovery, University of Wisconsin-Madison, Madison, WI, USA. [5] School of Pharmacy and Department of Chemistry, University of Wisconsin-Madison, Madison, WI, USA. [6] Department of Chemistry, Biomolecular Chemistry and Genome Center of Wisconsin, University of Wisconsin-Madison, Madison, WI, USA. [7] Morgridge Institute for Research, Madison, WI, USA. [8] Geriatric Research Education Clinical Center, Veterans Affairs Medical Center, Madison, Wisconsin, WI, USA. [9] Department of Neuroscience, University of Wisconsin-Madison, Madison, WI, USA. [10]Present address: Harvard Medical School, Boston, MA, USA. [11]Present address: Harvard T.H. Chan School of Public Health, Boston, MA, USA. [12]Present address: Fred Hutchinson Cancer Research Center, Seattle, WA, USA. [13]These authors contributed equally: Alexis J. Lawton, Yajing Peng, Qing Yu. Correspondence and requests for materials should be addressed to L.P. (email: lp1@medicine.wisc.edu)

The acetyl-CoA transporter, AT-1 (also referred to as SLC33A1), is a key member of the endoplasmic reticulum (ER) acetylation machinery, transporting acetyl-CoA from the cytosol into the ER lumen where acetyl-CoA serves as the acetyl-group donor for Nε-lysine acetylation[1,2]. Gene duplications in *AT-1/SLC33A1* have been identified in patients with autistic-like features, intellectual disability, and dysmorphic features; heterozygous mutations in *AT-1/SLC33A1* are associated with a familial form of spastic paraplegia, while homozygous mutations are associated with developmental delay and premature death[3–7]. Mouse models of altered AT-1 expression are effective models of human AT-1-linked diseases[8–10]. Both cell-based and mouse-based experiments support the conclusion that AT-1 activity regulates ER proteostasis by maintaining the balance between quality control and the induction of reticulophagy[1,2,8–14].

In this study, we investigate the outcomes of dysregulated AT-1 activity on intracellular acetyl-CoA homeostasis. Changes in the intracellular acetyl-CoA flux caused by hypoactive or hyperactive AT-1 could conceivably influence phenotypes beyond the secretory pathway; in particular the metabolic effects of these genetic manipulations have yet to be defined. For this purpose, we examine the hepatic molecular signatures of AT-1[S113R/+] mice[8], a model of AT-1 haploinsufficiency, and AT-1 sTg mice[10], a model of global AT-1 overexpression. The animals display distinct metabolic reprogramming across several intracellular compartments and pathways that is achieved through specific changes in both the proteome and the acetyl-proteome (protein acetylation). When taken together, our results suggest that AT-1 is an essential component of an intracellular communication network that promotes functional crosstalk between different cellular compartments and organelles to maintain acetyl-CoA homeostasis.

## Results

**Aberrant AT-1 activity alters lipid metabolism.** AT-1[S113R/+] mice are haploinsufficient for AT-1 activity (Fig. 1a), resulting in a ~50% decrease in acetyl-CoA transport from the cytosol into the ER lumen[8]. Histological assessment revealed that AT-1[S113R/+] mice have increased propensity to liver steatosis, which could be documented by both hematoxylin and eosin (H&E) (Fig. 1b) and electron microscopy (Fig. 1c). Although observed with both regular (40.2% carbohydrate, 4.7% fat) and breeder (44.7% carbohydrate, 9% fat) diets, the steatosis was much more common in animals fed the breeder diet. Importantly, the animals displayed no differences in food intake or body weight (Fig. 1d, e). Biochemical analysis revealed a ~60% increase in free acetyl-CoA available within the cytosol and a marked accumulation of fatty acids and triglycerides in the liver (Table 1). Finally, the serum lipid profiles were not different between wild-type (WT) and AT-1[S113R/+] mice (Table 2), indicating that the steatosis was not associated with dyslipidemia.

Contrary to AT-1[S113R/+], AT-1 sTg mice that systemically overexpress AT-1 had increased transport of acetyl-CoA from the cytosol to the ER lumen (Fig. 1f)[10]. If indeed the propensity of AT-1[S113R/+] to develop steatosis is caused by increased acetyl-CoA availability in the cytosol, then AT-1 sTg mice should be resistant, even when challenged with a lipogenic high-carbohydrate diet (HCD). To test this, both WT and AT-1 sTg animals were fed a HCD (70% carbohydrates, 5.2% fat) for 4 weeks[15,16]. As expected, WT mice on the HCD showed diffuse steatosis and lipid accumulation in the liver sections (Fig. 1g, h); however, AT-1 sTg on the HCD showed normal parenchyma and no histological evidence of lipid accumulation (Fig. 1g, h). We previously reported that AT-1 sTg eat more than their WT littermates[10], and although similar behavior was observed over

the 4 weeks of HCD feeding (Fig. 1i), there was no increase in body weight (Fig. 1j), suggesting that the changes in lipid metabolism were not due to altered food intake. When on a regular diet, AT-1 sTg mice showed reduced levels of cytosolic acetyl-CoA, fatty acids, and triglycerides as compared with WT littermates (Table 3). On the HCD, WT mice displayed a marked increase in fatty acids and triglycerides, whereas AT-1 sTg remained overall resistant to the lipogenic challenge (Table 3). Although there was a modest increase in triglycerides, the levels remained well below those of their WT littermates (Table 3). Furthermore, WT mice responded to the HCD by increasing serum triglyceride levels, while AT-1 sTg mice did not (Table 4).

An increase in oxidation of fatty acids is one possible explanation for the hepatic lipid-storage-resistance phenotype of AT-1 sTg mice. Consistent with this, fatty acid oxidative capacity was greater in primary hepatocytes from AT-1 sTg mice compared with WT controls (Fig. 1k). When taken together, the results obtained with AT-1[S113R/+] and AT-1 sTg mice indicate that changes in AT-1 activity and acetyl-CoA flux from the cytosol to the ER lumen can cause important changes in the availability of acetyl-CoA within the cytosol, impacting lipid metabolism and propensity to hepatic steatosis.

**Aberrant AT-1 activity affects the proteome.** To dissect the mechanism(s) underlying the above metabolic adaptation, we used quantitative mass spectrometry coupled with liquid chromatography and investigated possible proteomic changes within the liver of AT-1 sTg and AT-1[S113R/+] animals[17,18]. We found 2056 and 373 expression levels of proteins that were significantly ($P < 0.05$, Fisher's method) altered in AT-1 sTg and AT-1[S113R/+] mice, respectively (Fig. 2a, see also Supplementary Data 1). The cumulative difference in the distribution indicated that there are more downregulated proteins in AT-1[S113R/+] (mean FC = −0.122) than in AT-1 sTg (mean FC = −0.060) (Fig. 2b). Collectively, 61 proteins were affected in both models of AT-1 dysregulation; of them, 22 are associated with the secretory pathway, 8 with the mitochondria, 7 with the nucleus, and 24 with the cytosol (Fig. 2c, d). Comparing the proteomic response in the two mouse models, opposing changes in abundance were detected for 14 of these proteins, while the remaining 47 changed in the same direction in both models (Fig. 2d).

To assess the functional consequences of these protein expression changes, we conducted Kyoto Encyclopedia of Genes and Genomes (KEGG) pathway analysis using the Over-Representation Analysis method. Enriched pathways passed a Benjamini–Hochberg test with a FDR < 0.05. Twenty-nine pathways were enriched in the AT-1 sTg, and 14 pathways were enriched in AT-1[S113R/+], of which 6 were overlapping (Fig. 2e). These KEGG pathways were then categorized into broad key pathways (Fig. 2f). In addition to secretory pathway and mitochondrial processes anticipated based on prior work in the neuron-specific mouse model of AT-1 overexpression[9], here, other pathways were also identified. Pathways responsive to changes in AT-1 included lipid metabolism, glucose metabolism, and amino acid metabolism. KEGG pathways enriched in both models of AT-1 dysregulation included carbon metabolism, fatty acid degradation, fatty acid metabolism, butanoate metabolism, PPAR signaling, and BCAA degradation. Despite both models perturbing the activity of AT-1, we observed unique protein expression consequences in each model. Specifically, AT-1[S113R/+] displayed overall more downregulated proteins, and distinct enrichment in many mitochondrial pathways and lipid metabolism pathways. In contrast, AT-1 sTg displayed a wider distribution of protein expression and a wider array of enriched pathways: transcription, translation, secretory pathway, lipid

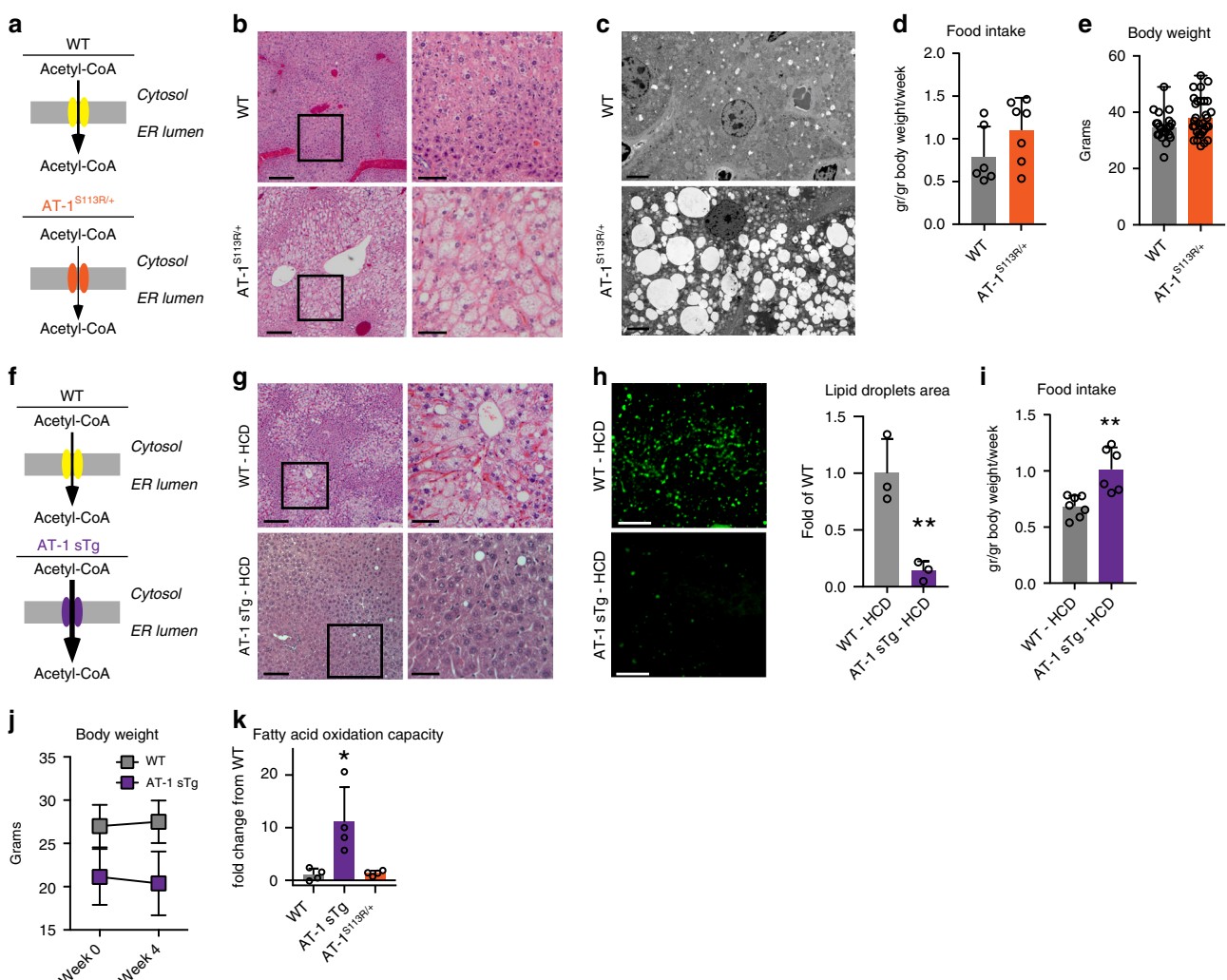

**Fig. 1** AT-1$^{S113R/+}$ show spontaneous steatosis while AT-1 sTg are resistant to diet-induced steatosis. **a** Schematic representation of reduced acetyl-CoA transport through AT-1 in the haploinsufficient AT-1$^{S113R/+}$ mice. **b** Representative H&E staining of liver sections from WT and AT-1$^{S113R/+}$ mice. Scale bar, 250 μm (left), 100 μm (right). **c** Representative electron microscopy of the liver sections from WT and AT-1$^{S113R/+}$ mice. Scale bar, 5 μm. **d** Food intake of AT-1$^{S113R/+}$ ($n = 6$) mice and age-matched WT ($n = 7$) littermates. **e** Body weight of 6-month-old AT-1$^{S113R/+}$ ($n = 31$) mice and age-matched WT ($n = 23$) littermates. **f** Schematic representation of increased acetyl-CoA transport through AT-1 in systemic overexpression AT-1 sTg mice. **g** Representative H&E staining of the liver sections from WT and AT-1 sTg fed a high-carb diet (HCD). Scale bar, 250 μm (left), 100 μm (right). **h** LipidTOX green staining of the liver sections of WT and AT-1 sTg mice fed a HCD. Representative images (left panel) and quantification (right panel) are shown (WT, $n = 3$; AT-1 sTg, $n = 3$). Scale bar, 25 μm. **i** Food intake of AT-1 sTg ($n = 6$) and WT ($n = 7$) littermates while fed a high-carb diet. **j** Body weight of 3-month-old AT-1 sTg ($n = 8$) mice and age-matched WT ($n = 8$) littermates at week 0 and week 4 of the high-carb diet. **k** Fatty acid oxidative (FAO) capacity as examined by Seahorse assay in primary cultured hepatocytes (hepatocytes from biologically independent animals; WT, $n = 4$; AT-1 sTg, $n = 4$; AT-1$^{S113R/+}$, $n = 4$). Statistical analysis was performed using a one-tailed Student's $t$ test. *$P < 0.05$; **$P < 0.005$; #$P < 0.0005$; All data are represented as mean ± s.d. Circles indicate individual animals

**Table 1 Hepatic lipid content in WT and AT-1$^{S113R/+}$ mice fed standard diet**

|  | WT | AT-1$^{S113R/+}$ |
|---|---|---|
| Cytosolic acetyl-CoA (nmol/g tissue) | 2.42 ± 0.97 | 3.93 ± 1.65 |
| Fatty acids (μmol/g tissue) | 18.80 ± 7.69 | 36.70 ± 5.21** |
| Triglycerides (μmol/g tissue) | 17.65 ± 4.60 | 67.01 ± 22.00* |
| Total cholesterol (μmol/g tissue) | 4.21 ± 0.87 | 5.44 ± 1.42 |

Hepatic lipid content of WT ($n = 5$) and AT-1$^{S113R/+}$ ($n = 5$) mice fed a standard diet. One-tailed Student's $t$ test. *$P < 0.05$; **$P < 0.005$

metabolism, glucose metabolism, mitochondria adaptation, and amino acid metabolism, indicating that the overexpression of AT-1 causes highly global protein expression changes. These data show that AT-1 responsive proteins and pathways extend to metabolic pathways not previously known to be affected by changes in the cytosol-to-ER acetyl-CoA flux.

**Aberrant AT-1 activity affects the acetyl-proteome**. Acetyl-CoA-dependent processes not only include fatty acid metabolism and the citric acid cycle but also Nε-lysine acetylation, which can occur in the nucleus and cytoplasm[19], ER[2,14], mitochondria[20], and the peroxisome[21]. To assess the ability of AT-1 and acetyl-CoA availability to modulate protein function through lysine acetylation, we used a mass spectrometry method that elucidates the stoichiometry of site-specific lysine acetylation within proteomes[22].

We quantified the steady-state acetylation stoichiometry on lysine sites at the proteome level, which ranged from < 1 to 99%. In the AT-1 sTg and AT-1$^{S113R/+}$ liver, 3341 and 3003 lysine sites were identified, respectively. When compared with WT, 375 were significantly changing in the AT-1 sTg, and 415 were significantly changing in the AT-1$^{S113R/+}$ (Fig. 3a; see also Supplementary Data 2). The distribution of stoichiometry changes (Fig. 3b) as well as the average number of lysine sites per protein identified (Fig. 3c) were significantly different between the two models (see also later). These differences can possibly be explained by changes in the availability of cytosolic acetyl-CoA as a result of less (in the AT-1$^{S113R/+}$ mice) or more (in the AT-1 sTg mice) AT-1-mediated transport into the ER lumen.

A protein with multiple lysine sites can display varying stoichiometry on each site, leading to multiple acetylated protein forms capable of influencing protein stability and activity. Comparison of the AT-1 models revealed a 12% (88 sites) overlap in significantly changing acetylation sites, and 21% (119 proteins) overlap in modified proteins (Fig. 3d). To assess the functional implications of these posttranslational modifications, we conducted the KEGG pathway analysis of the acetyl-proteome.

We found a compelling 57% overlap in enriched pathways in both models of AT-1 dysregulation, which include lipid metabolism, amino acid metabolism, mitochondria, and secretory pathway and proteostasis (Fig. 3e).

The 88 acetylation sites that were detected in both models are localized as follows: 26 in the secretory pathway, 18 in the mitochondria, 33 in the cytosol, and 11 in the nucleus. Strikingly, 85 of the acetylation sites had increased acetylation stoichiometry in one mouse model and decreased acetylation stoichiometry in the other model, whereas only 3 acetylation sites had stoichiometry changes in the same direction in both models compared with WT (Fig. 3f). With the exception of eight acetylated sites, all of the lysine sites in the AT-1$^{S113R/+}$ showed hyperacetylation, whereas the lysine site in the AT-1 sTg showed hypoacetylation. Lastly, we evaluated the subcellular location of all acetylated proteins significantly different from WT; the general trends again indicate hypoacetylation in AT-1 sTg and hyperacetylation in AT-1$^{S113R/+}$ (Fig. 3g). When broken down by compartment, this distribution was again maintained across the secretory pathway, mitochondria, nucleus and cytosol (Fig. 3h). This differential response likely reflects changes in substrate availability. Indeed, the increased transport of cytosolic acetyl-CoA into the ER lumen in AT-1 sTg mice reduced acetyl-CoA availability in the cytosol, whereas the reduced transport to the ER in AT-1$^{S113R/+}$ mice had the opposite effect. These results strongly suggest that these coordinated posttranslational acetyl modifications are intimately linked to the levels of cytosolic acetyl-CoA, and that AT-1 activity is a major driver of protein acetylation in these mouse models.

**AT-1 controls cell metabolism and mitochondria bioenergetics**. Comparison of the proteomic and acetyl-stoichiometric data indicate that the metabolic reprogramming induced by changes in AT-1 activity and acetyl-CoA availability is quite complex with proteins being regulated either at the level of the proteome or acetylome, or at both levels (Fig. 4a). Comparison of the two data sets demonstrated that there are 105 and 35 acetylation sites in the AT-1 sTg and AT-1$^{S113R/+}$ models, respectively, that are significantly changed at both the protein and posttranslational (acetylation) level as compared with WT.

To further understand how these different modes of regulation can integrate into specific metabolic responses, we

---

**Table 2 Lipid serum profile of WT and AT-1$^{S113R/+}$ mice fed standard diet**

|  | WT | AT-1$^{S113R/+}$ |
|---|---|---|
| Total cholesterol (mg/dl) | 142.0 ± 24.8 | 143.6 ± 2.8 |
| Triglycerides (mg/dl) | 200.7 ± 69.4 | 224.3 ± 52.2 |

Lipid serum profile of WT ($n = 6$) and AT-1$^{S113R/+}$ ($n = 6$) mice fed a standard diet. Statistical analysis was performed using a one-tailed Student's $t$ test

---

**Table 3 Hepatic lipid content in WT and AT-1sTg mice fed control and high-carb diet**

|  | Control diet | | High-carb diet | |
|---|---|---|---|---|
|  | WT | AT-1 sTg | WT | AT-1 sTg |
| Cytosolic acetyl-CoA (nmol/g tissue) | 5.02 ± 1.83 | 3.19 ± 0.33* | 5.70 ± 2.91 | 4.88 ± 2.83 |
| Fatty acids (μmol/g tissue) | 9.28 ± 2.45 | 6.27 ± 1.40** | **15.99 ± 5.54**** | 6.36 ± 3.48** |
| Triglycerides (μmol/g tissue) | 20.01 ± 7.79 | 6.92 ± 2.03# | **34.38 ± 7.97**# | **10.65 ± 4.49**#* |
| Total cholesterol (μmol/g tissue) | 3.26 ± 0.64 | 3.31 ± 0.42 | 4.17 ± 1.25 | 4.04 ± 0.52 |

Hepatic lipid content of WT and AT-1 sTg mice (control diet, $n = 5$ per group; HCD, $n = 5$ per group). Bold indicates difference between diet groups, non-bold indicates difference between WT and AT-1 sTg mice on the same diet. One-tailed Student's $t$ test. *$P < 0.05$; **$P < 0.005$; #$P < 0.0005$

---

**Table 4 Lipid serum profile in WT and AT-1 sTg mice fed control and high-carb diet**

|  | Control Diet | | High-Carb Diet | |
|---|---|---|---|---|
|  | WT | AT-1 sTg | WT | AT-1 sTg |
| Total cholesterol (mg/dl) | 119.2 ± 19.3 | 60.6 ± 9.5# | 110.1 ± 44.7 | 61.8 ± 38.9* |
| Triglycerides (mg/dl) | 72.2 ± 8.3 | 67.8 ± 21.3 | **117.1 ± 47.4*** | 84.2 ± 24.7 |

Lipid serum profile of WT and AT-1 sTg mice (control diet, $n = 5$ per group; HCD, $n = 5$ per group). Bold indicates difference between diet groups, non-bold indicates difference between WT and AT-1 sTg mice on the same diet. One-tailed Student's $t$ test. *$P < 0.05$; #$P < 0.0005$

---

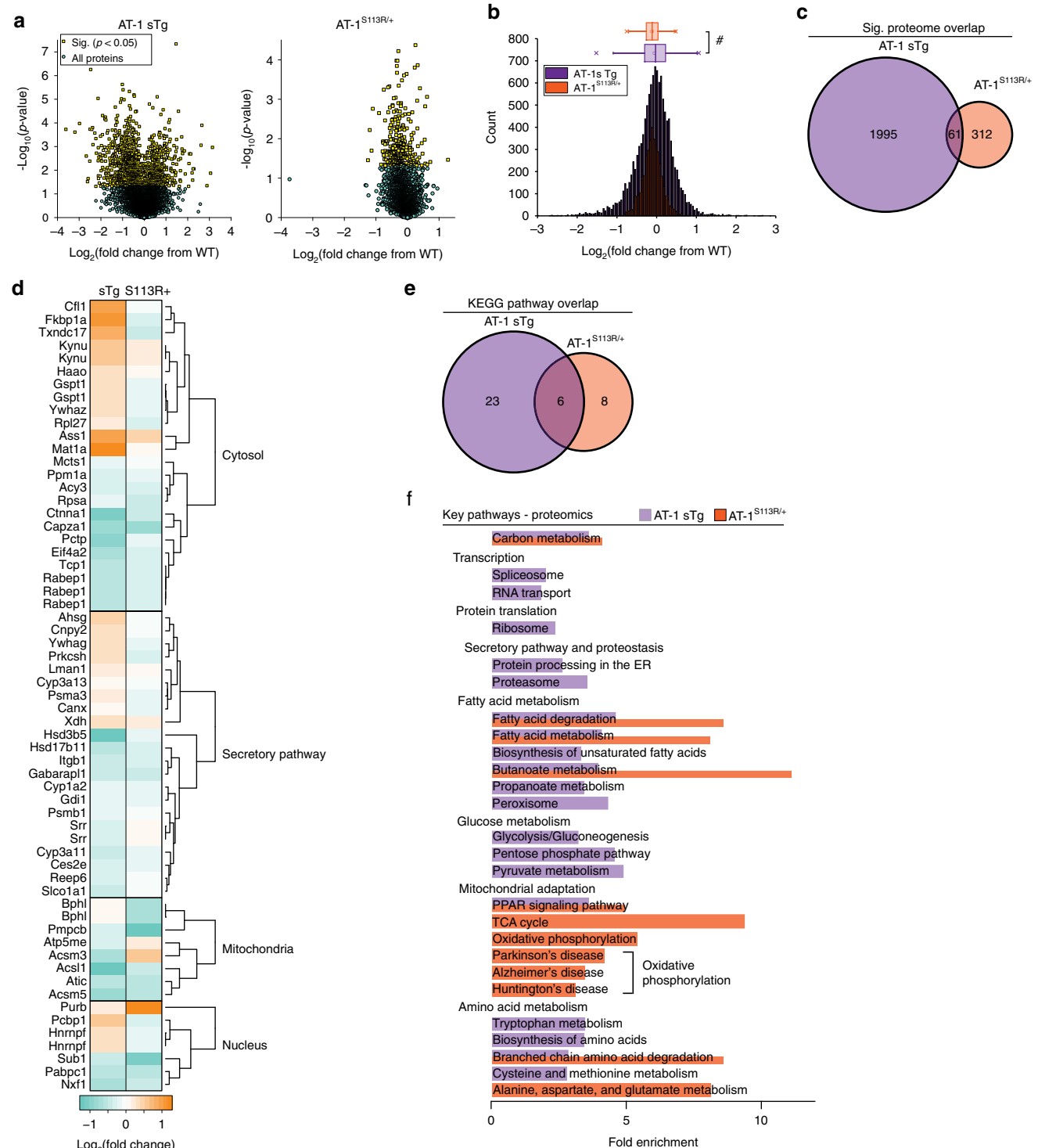

focused on the two metabolic clusters which are immediately relevant to the liver phenotype (see Fig. 1) and were heavily highlighted in the pathway analysis—lipid and mitochondrial metabolism. Proteins that were identified in the enriched KEGG pathways from either model at the level of the proteome and acetyl-proteome were included in the cluster analysis. Factors regulated at the level of the proteome and/or acetylome were represented within each cluster (Fig. 4b, c; see also Supplementary Tables 1, 2). Again, these results reveal the far-reaching metabolic impact of changes in AT-1 activity and consequent acetyl-CoA availability.

An important and unexpected finding of this study is the fact that the mitochondria dynamically adapt to changes in acetyl-CoA transport from the cytosol to the ER lumen imparted by the activity of the ER membrane acetyl-CoA transporter, AT-1. Also unexpected is that fact that the mitochondria adaptation can involve changes in stoichiometry of lysine acetylation that was expected to be buffered from acetyl-CoA levels in the cytosol. In line with this finding, we investigated the functional significance of the mitochondria adaptive response.

Direct visualization of the mitochondria by structure-illumination microscopy (SIM) in primary cultured hepatocytes

**Fig. 2** AT-1 sTg and AT-1[S113R/+] display proteomic changes across many metabolic pathways. **a** Volcano plot displaying all quantified proteins in AT-1 sTg ($n = 4$; left) and AT-1[S113R/+] ($n = 4$; right) livers, compared with age-matched WT littermates. Statistically significant proteins (2056 in AT-1 sTg; 373 in AT-1[S113R/+]) are highlighted in yellow, and all other proteins are designated in blue. Fisher's method, $P < 0.05$. **b** Histogram and box and whiskers plot showing the distribution of all proteins' $\log_2$ fold change in AT-1 sTg and AT-1[S113R/+], as compared with WT. The data represented as box and whisker plots: box represents the 25th to 75th inner quartile range, with the middle line denoting the median, and the inner square denoting the mean; the whiskers represent the interquartile distance with a coefficient of 1.5; the $x$ represents the 1st and 99th percentile. Kolmogorov–Smirnov test. #$P < 0.0005$. **c** Venn diagram displaying significant protein overlap within models of AT-1 dysregulation. All proteins shown in panel **a** (yellow) were analyzed. **d** Heat-map showing the expression profile of the 61 overlapping proteins found in both AT-1 sTg and AT-1[S113R/+]. Subcellular localization was determined according to the Uniprot database annotation. Secretory pathway includes the endoplasmic reticulum, Golgi apparatus, proteasome, lysosome, extracellular, and secreted proteins. Cytoplasm includes cytoskeleton, cytosol, ribosomes, and all other organelles. **e** Venn diagram displaying KEGG pathways enriched in the significant proteins within models of AT-1 dysregulation. Parameters for enrichment can be found in the proteomics methods. **f** The fold enrichment of KEGG pathways were organized by key cellular processes within models of AT-1 dysregulation. The original data set is shown in Supplementary Data 1. All proteins analyzed in **c–f** were significantly different ($P < 0.05$, Fisher's method) from age-matched WT littermates

revealed significant expansion of the mitochondria network in AT-1 sTg mice (Fig. 4d), with significant increases detected in both mitochondrial area and volume (Fig. 4e). Mitochondrial remodeling was associated with a significant upregulation of the mRNA levels of the transcriptional co-activator *peroxisome proliferator-activated receptor-γ (PPAR-γ) co-activator 1α (PGC-1α)* (Fig. 4f), which is an important regulator of mitochondrial biogenesis and mitochondrial function[23,24]. These results are in line with the expansion of PPAR signaling observed at the level of the proteome (see Fig. 2; also discussed later). Next, we used U-[13]C glutamine isotope tracing to measure metabolites in primary hepatocytes and determine the functional significance of the TCA/citric acid cycle adaptation identified in the proteome and acetyl stoichiometry data. The results revealed increased levels of [13]C-labeled alpha-ketoglutarate and fumarate, increased total levels of alpha-ketoglutarate, and a trend ($P = 0.055$, Student's $t$ test) toward increased malate levels in AT-1[S113R/+] mice (Fig. 4g; see also Supplementary Data 3 and Supplementary Table 3). No significant differences were observed in TCA/citric acid metabolites in AT-1 sTg animals.

Together, data shown here suggest that AT-1 sTg and AT-1[S113R/+] mice are adapting in a different capacity to AT-1 activity and acetyl-CoA availability. In AT-1 sTg mice, we detected proteomic enrichment in PPAR signaling and acetylated protein enrichment in the TCA cycle. These changes were reflected functionally in an overall expansion in the mitochondria, but without a change in overall levels of TCA metabolites. In contrast, in AT-1[S113R/+] mice, we detected an increase in engagement of the TCA/citric acid cycle, which was reflected in the proteomic pathway enrichment of PPAR signaling, TCA cycle, and oxidative phosphorylation and acetyl proteoform enrichment in PPAR signaling and the TCA cycle.

The increased fatty acid oxidation capacity observed in AT-1 sTg mice (see Fig. 1k) is also reflected in the expansion of the mitochondria network (Fig. 4d) and the adaptive response of the carnitine-acylcarnitine translocase machinery (Slc25a20, Cpt1a and Cpt2) and the hydroxyacyl-CoA dehydrogenase/3-ketoacyl-CoA thiolase/enoyl-CoA hydratase trifunctional enzyme (Hadh, Hadha, and Hadhb). In fact, we found Hadhb, Cpt2, and Cpt1a to be modified at the proteome level, Hadh and Hadha at the acetylome level, and Slc25a20 both at the proteome and acetylome level (Fig. 4b, c; see also Supplementary Tables 1, 2). Together, these adaptive changes at the proteome and acetylome level integrate to promote the oxidation of fatty acids within the mitochondria.

## Discussion

Cell-based and mouse-based studies support the conclusion that AT-1 activity regulates ER proteostasis by maintaining the balance between quality control and the induction of reticulophagy[1,2,8–14]. However, changes in intracellular acetyl-CoA flux caused by hypoactive or hyperactive AT-1 could conceivably influence cellular events beyond those already established; in particular the metabolic effects of these genetic manipulations have not been defined.

In this study, we used two models of AT-1 dysregulation and altered acetyl-CoA flux: AT-1[S113R/+] mice, a model of AT-1 haploinsufficiency, and AT-1 sTg mice, a model of AT-1 over-expression. We discovered that AT-1 activity has pleiotropic effects that go beyond quality control and the induction of reticulophagy. Specifically, the animals display distinct metabolic adaptation across intracellular compartments, including reprogramming of lipid metabolism and mitochondria bioenergetics. Phenotypically, this adaptive response in the AT-1[S113R/+] model leads to spontaneous steatosis and increased engagement of the TCA cycle. In contrast, in the AT-1 sTg model, it leads to resistance to diet-induced steatosis and expansion of the mitochondria network. Mechanistically, this functional adaptation is achieved by global reprograming of several biological pathways caused by specific changes in both the proteome and the acetyl-proteome (protein acetylation). Importantly, the proteome and acetyl-proteome of AT-1 mouse models support these cellular phenotypes—also showing dramatic changes across lipid metabolism and mitochondria-related pathways. Therefore, AT-1 has emerged as a key regulator in intracellular acetyl-CoA homeostasis, which has far reaching consequences within the cell's metabolism.

To maintain homeostasis, the cell must ensure crosstalk between different organelles and compartments. It seems likely that key cellular metabolites reflect the immediate activity of metabolic enzymes as well as the functional metabolic state of intracellular organelles. In this way, signaling molecules and/or availability respond to extracellular and/or intracellular changes and allow for implementation of the appropriate adaptive response. Our results indicate that changes in acetyl-CoA flux from the cytosol to the ER lumen, as caused by reduced or increased AT-1 activity, cause significant metabolic adaptation. These results suggest that the cytosol-to-ER flux of acetyl-CoA is a branch of the more general nutrient-signaling-pathway[2,25–27] that enables rapid modulation and reprogramming of different intracellular activities upon fluctuation of metabolites/nutrients.

Mechanistically, AT-1 responsive proteins are involved in the secretory pathway, mitochondria, lipid, glucose, and amino acid metabolism. The AT-1 response includes changes in acetyl proteoforms in a manner that appears to be highly coordinated and occurs within different cellular organelles and compartments. These data suggest that acetyl-CoA not only serves as a cytosolic sensor but that the status of cytosol-to-ER acetyl-CoA flux is an essential component of an intracellular communication network

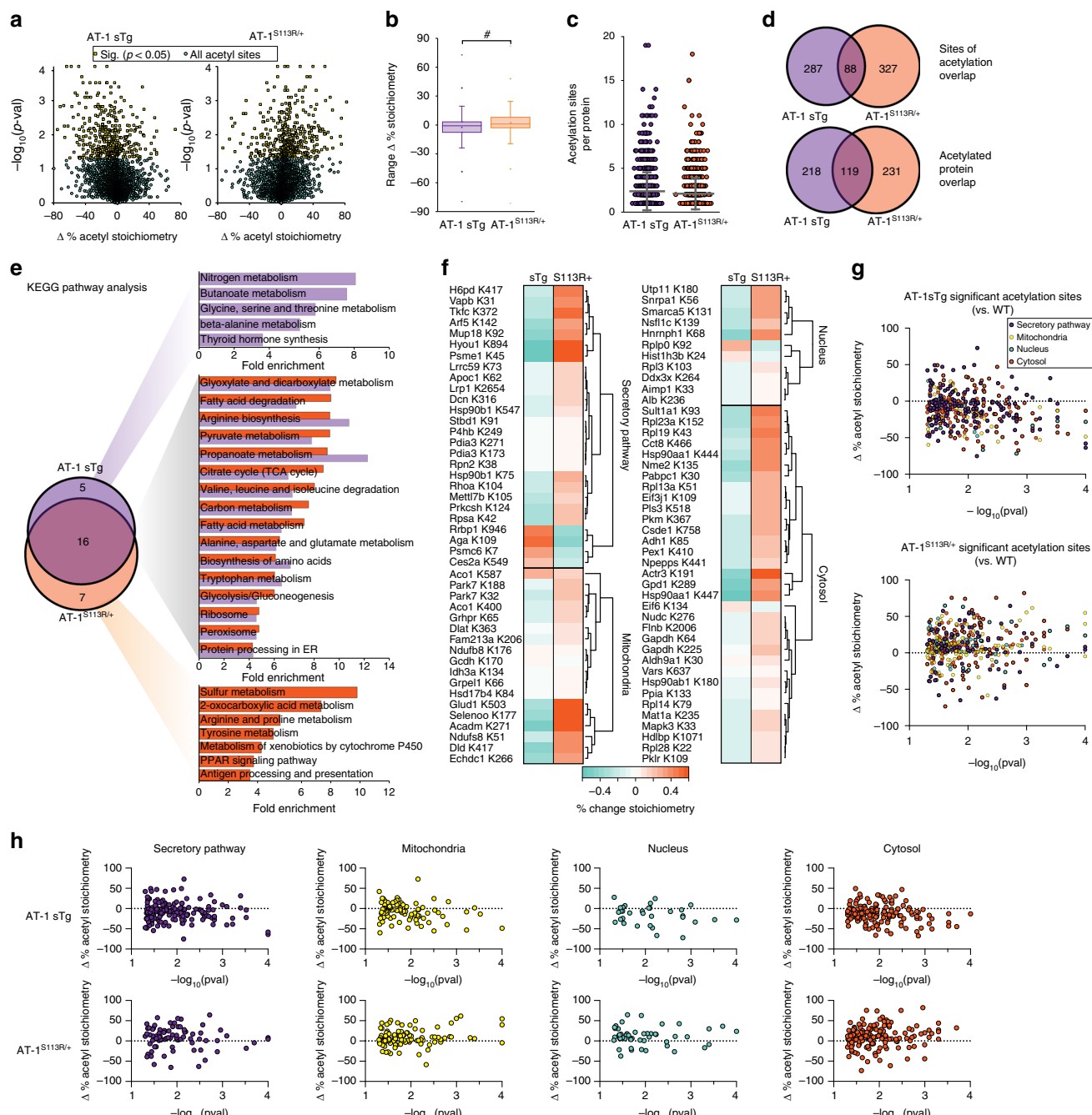

**Fig. 3** AT-1 sTg and AT-1$^{S113R/+}$ mice show changes in stoichiometry of lysine acetylation across many metabolic pathways. **a** Volcano plot of percent change in lysine acetylation sites detected in AT-1 sTg ($n = 4$; left) and AT-1$^{S113R/+}$ ($n = 4$; right) mice, compared with age-matched WT littermates. Statistically significant sites (375 in AT-1 sTg; 415 in AT-1$^{S113R/+}$) are highlighted in yellow, and all other detected sites are designated in blue. One-way ANOVA, $P < 0.05$. **b** Percent change of lysine site stoichiometry of all detected lysine sites in both models of AT-1 dysregulation. The data represented as box and whisker plots: box represents 25th to 75th inner quartile range; middle line denoting the median, and the inner square denoting the mean; whiskers represent the interquartile distance with a coefficient of 1.5; the most extreme values are the minimum and maximum, and the other dots are the 99th percentile and 1st percentile. Kolmogorov–Smirnov test. #$P < 0.0005$. **c** The number of acetylation sites affected per protein. **d** Venn diagram showing the overlap between significant lysine sites (upper) and the overlap between acetylated proteins (lower) in AT-1 sTg and AT-1$^{S113R/+}$ mice. **e** Venn diagram of enriched KEGG pathways, and their enrichment scores found in models of AT-1 dysregulation. Purple denotes AT-1 sTg, and orange denotes AT-1$^{S113R/+}$. Parameters for enrichment can be found in the Methods section. **f** Heatmap showing the percent change in stoichiometry of the 88 overlapping sites found in both models. Subcellular localization was determined according to the Uniprot database annotation. The secretory pathway includes the endoplasmic reticulum, Golgi apparatus, proteasome, lysosome, extracellular, and secreted proteins. Cytoplasm includes the cytoskeleton, cytosol, ribosomes, and all other organelles. **g, h** The distribution of significant lysine acetylation sites according to their Uniprot subcellular annotation, as a global view in AT-1 sTg (upper) and AT-1$^{S113R/+}$ (lower) mice (**g**), and arranged by localization: the secretory pathway (purple circles), mitochondria (yellow circles), nucleus (blue circles), and cytosol (orange circles) (**h**). The original data set is shown in Supplementary Data 2. All sites analyzed in **d**–**h** were significantly different ($P < 0.05$, one-way ANOVA) from age-matched WT littermates. Bars are mean ± s.d. Circles indicate the average value of each acetylation site

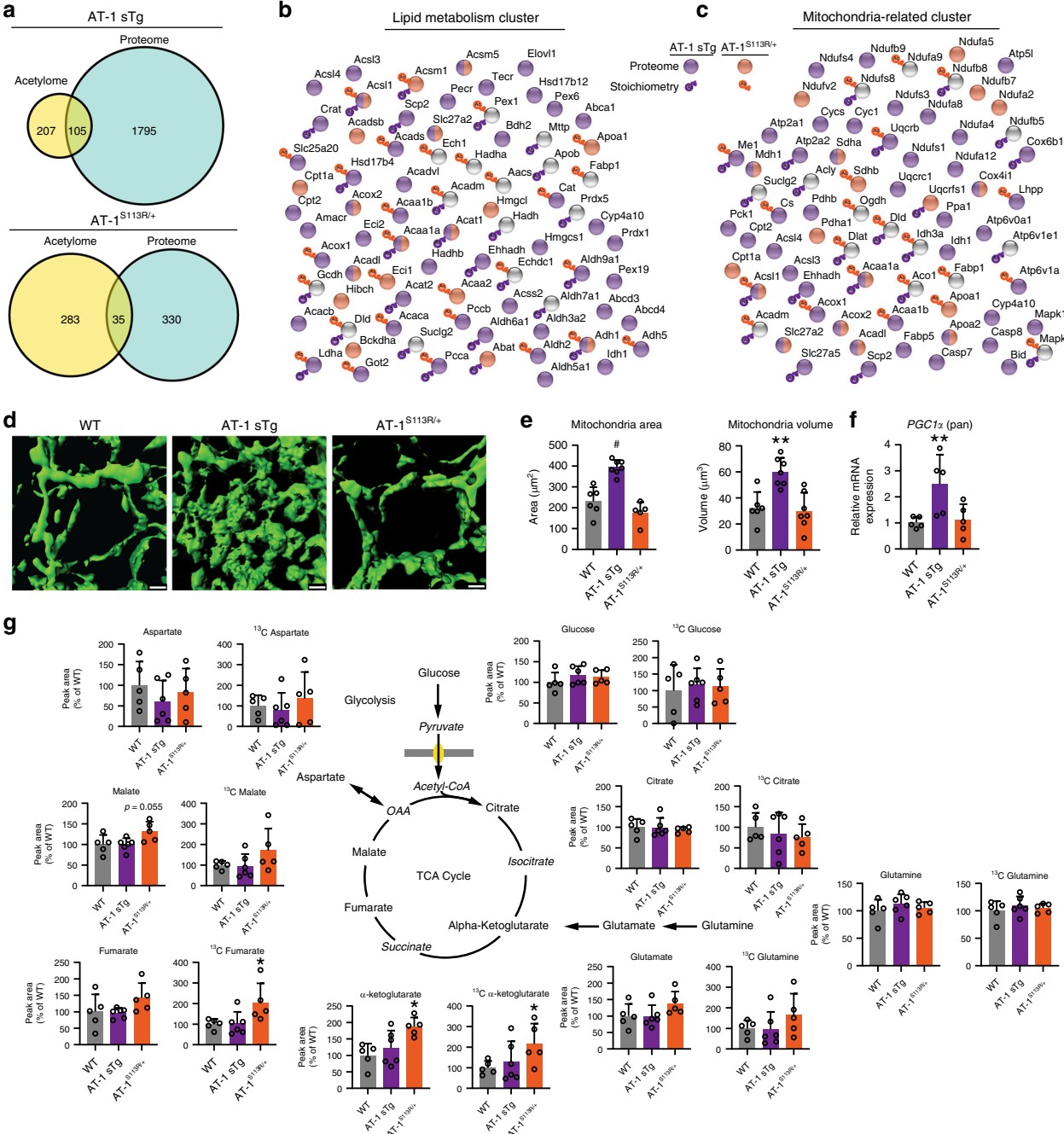

**Fig. 4** AT-1 sTg and AT-1$^{S113R/+}$ mice show significant metabolic and mitochondria adaptation. **a** Venn diagram of overlapping significant proteins found to be modified in their expression levels (proteome) and in their stoichiometry of lysine acetylation (acetylome) within AT-1 sTg (upper) and AT-1$^{S113R/+}$ (lower) livers. **b**, **c** Proteins found to be enriched in lipid metabolism (**b**) and mitochondria-related (**c**) KEGG pathways are shown in clusters according to their level of regulation. Changes in the stoichiometry of acetylation are indicated by a purple (AT-1 sTg) or orange (AT-1$^{S113R/+}$) acetyl group; changes in the proteome are indicated by a purple (AT-1 sTg) or orange (AT-1$^{S113R/+}$) circle; no change in proteome level is indicated by a gray circle. **d**, **e** Mitochondria morphology in primary-cultured hepatocytes was examined with CellLight mitochondria stain (scale bar, 1 μm) (**d**) and quantified using Imaris reconstruction of area and volume (**e**) (hepatocytes from biologically independent animals; WT, n = 6; AT-1 sTg, n = 7 ; AT-1$^{S113R/+}$, n = 7). One-tailed Student's t test. \*\*P < 0.005; #P <0.0005. **f** mRNA levels of PGC1α (pan) in liver samples (livers from biologically independent animals; WT, n = 5; AT-1 sTg, n = 5; AT-1$^{S113R/+}$, n = 5). One-tailed Student's t test. \*\*P < 0.005. **g** Primary cultured hepatocytes were labeled with $^{13}$C-glutamine for 30 min before collection. The total TCA intermediate metabolite levels and $^{13}$C-labeled metabolite levels are shown (hepatocytes from biologically independent animals; WT, n = 5; AT-1 sTg, n = 6; AT-1$^{S113R/+}$, n = 5). Two-tailed Student's t test. \*P < 0.05. All data are represented as mean ± s.d.

in place to maintain acetyl-CoA homeostasis. Overall, the results support the conclusion that the flux of acetyl-CoA from the cytosol to the ER lumen can act as a metabolic regulator, and directly impacts crosstalk between different intracellular organelles and compartments.

## Methods

**Animals.** All animal experiments were carried out in accordance with the NIH Guide for the Care and Use of Laboratory Animals and received ethical approval by the Institutional Animal Care and Use Committee of the University of Wisconsin-Madison. Generation of AT-1$^{S113R/+}$ animals was achieved by crossing mice carrying the *Slc33a1*-S113R mutation to WT animals[8]. AT-1$^{S113R/+}$ mice were studied

between 5 and 7 months of age. Generation of AT-1 sTg mouse was achieved by crossing Rosa26:tTA mice with pTRE AT-1 mice to generate ROSA26:tTA;pTRE-AT-1 (AT-1 sTg mice)[10]. AT-1 sTg mice were studied at ~3 months of age. Age-matched wild-type (WT) littermates were used as controls. Males were used for the experiments in this study.

**Diets**. The following diets were used throughout our study: regular chow, teklad 8604; breeder chow, teklad 2019; high-carbohydrate diet (HCD), teklad 98090.

**Hepatocyte isolation**. Primary mouse hepatocytes were prepared by anesthetizing the animals by intraperitoneal injection of 0.4 mg/g freshly prepared 2, 2, 2-tribromoethanol (Avertin). The abdominal cavity was opened and a cannula was inserted into the inferior vena cava. The flow of perfusion buffer (HBSS, 59 mM HEPES, 0.6 mM EGTA; pH 7.5) began, and the portal vein was immediately cut. The liver was then perfused with collagenase buffer (HBSS, 0.25 mg/ml of collagenase type IV (Sigma-Aldrich C5138), 7 mM $CaCl_2$, 53 mM HEPES, pH 7.5). Next, the liver was excised, and cells were harvested for tissue culture experiments by cutting into the liver capsule. Hepatocytes were passed through a 70 -µm cell strainer and centrifuged ($100 \times g$, 5 min), re-suspended in washing media (M199, 1× glutamax), centrifuged again ($100 \times g$, 5 min), re-suspended in plating media (10% FBS, 1× glutamax, 1× penicillin/streptomycin/glutamine, 1× BSA) and plated. For immunocytochemistry, hepatocytes were plated at 100,000 cells per collagen-coated coverslip (GC-12, Neuvitro), and transfected with CellLight Mitochondria-GFP BacMam 2.0 (ThermoFisher Scientific) overnight. Transfected hepatocytes were fixed with paraformaldehyde (4%, 15710, Electron Microscopy Sciences) for 10 min, followed by permeabilization with 0.1% Triton-X100 (Roche Applied Science) for 5 min and incubation in blocking buffer (10% BSA, 5% goat serum in PBS) for one hour. Nuclei were stained with DAPI (62248, ThermoFisher Scientific) during the blocking step. Cells were washed three times in PBS and prolong diamond antifade mountant (P36965, ThermoFisher Scientific) was used to mount the coverslips. Images were acquired using Structured Illumination Microscopy (Nikon SIM), and analyzed in Imaris imaging software (Bitplane, Oxford Instruments) using the Surfaces module.

**Lipid determinations**. Whole liver samples were homogenized in PBS and the following assays were performed: Coenzyme A was measured using the CoA assay kit (K367, BioVision); free fatty acid was measured using the free fatty acid assay kit (ab65341, Abcam); triglycerides were measured using the triglyceride assay kit (ab65336, Abcam); the total cholesterol was measured using the cholesterol assay kit (ab65359, Abcam). Cytosolic acetyl-CoA was isolated from the total liver, and was measured using the acetyl-CoA assay kit (ab87546, Abcam). To isolate the cytosol, the liver was homogenized (in 10 mM HEPES, 10 mM KCl, 0.1 mM EDTA, 0.1 mM EGTA, 1 mM DTT, protease and phosphatase inhibitors), then centrifuged ($600 \times g$, 10 min). The supernatant was collected and centrifuged again ($89,300 \times g$, 90 min) for analysis. For lipid serum analytes, blood was collected transcardially from mice with an insulin syringe. Blood was clotted on ice for 15 min, then centrifuged at $200 \times g$ for 10 min at 4 °C and the supernatant was collected. The lipid serum profile was performed by the University of Wisconsin-Madison Clinical laboratory.

**Real-time PCR**. Real-time PCR was performed using the Roche 480 lightcycler and Sybr Green Real Time PCR Master Mix (Life Technologies). Gene expression levels were normalized against GAPDH levels and expressed as fold of change from control. The cycling parameters for PGC1-α Pan (all PGC1-α isoforms) were as follows: 95 °C, 15 s; 60 °C, 60 s, for a maximum for 40 cycles. Primer sets were as follows: *PGC-1α (pan)*: 5′-TGATGTGAATGACTTGGATACAGACA-3′ (sense), 5′-GCTCATTGTTGTACTGGTTGGATATG-3′ (antisense); GAPDH: 5′-AGGTC GGTGTGAACGGATTTG-3′(sense), 5′-TGTAGACCATGTAGTTGAGGTCA-3′ (antisense).

**Histology**. Transmission electron microscopy was performed at the Electron Microscopy facility of the William S. Middleton Memorial Veterans Hospital (Madison, WI). Tissue sections were observed using the transmission electron microscopy (H-600; Hitachi) at 75 kV. For LipidTox staining, liver sections were collected and immediately placed in the Optimal Cutting Temperature medium (ThermoFisher Scientific, San Jose, CA, USA), and stored at −80 °C until processed. Sections were sliced on a microtome cryostat (Microm HM 505 N), and processed using HCS LipidTOX™ Neutral Lipid Stains protocol (ThermoFisher Scientific, San Jose, CA, USA). Processed slides were imaged on a macroconfocal microscope (Leica TCS-LSI). For H&E staining, the liver sections were collected and immediately placed in 10% neutral buffered formalin overnight. Sections were then embedded in paraffin using standard techniques. Sections were sliced on a microtome; they underwent deparaffinization and rehydration, and processed for H&E staining. Processed slides were imaged on a Zeiss Axiovert 200 inverted fluorescent microscope.

**Fatty acid oxidation**. Mitochondria fuel oxidation was determined using a Seahorse XF mitochondria Fuel Flex Test kit (Agilent Technologies, Wilmington, DE, USA). Primary hepatocytes from mice were plated at $1.0 \times 10^4$ cells/well on a 96-well collagen-coated microplate overnight. One hour before the assay, the media was changed to 180 µL/well of XF base media (Seahorse XF DMEM, 1 mM

pyruvate, 2 mM glutamine, 10 mM glucose; pH 7.4) and placed at 37 °C non-$CO_2$ incubator for 1 h. The following three fuel pathway inhibitors were added during the assay: UK5099 (to inhibit glucose oxidation), Etomoxir (to inhibit long chain fatty acid oxidation), and BPTES (to inhibit glutamine oxidation). Fatty acid oxidation capacity was tested by injecting UK5099 (2.0 µM final) and BPTES (3.0 µM final) after the third measurement of baseline oxygen consumption. Six rate measurements proceeding this injection, Etomoxir (4.0 µM final) was injected. The equation for FAO capacity was [1-((baseline OCR-UK5099/BPTES OCR)/(baseline OCR – all inhibitors OCR))] and was expressed as a function of cellular oxygen consumption rate.

**Subcellular fractionation**. In order to improve our ability to capture changes at the subcellular compartment level, the liver was processed through a crude fractionation protocol prior to quantitative proteomics and stoichiometry of acetylation. The liver was homogenized in Buffer A (10 mM Tris-HCl, 10 mM NaCl, 3 mM $MgCl_2$; pH 7.4) and passed through a 70-µm cell strainer before centrifugation ($900 \times g$, 5 min). The pellet was washed in PBS and centrifuged ($900 \times g$, 5 min), twice and retained. For the cytosolic and mitochondrial fraction, the original supernatant was centrifuged again ($21,000 \times g$, 2 min); supernatant was retained and re-centrifuged and the final supernatant contained the cytosolic fraction. The pellet containing the mitochondria was resuspended in Buffer B (10 mM Tris-HCl, 10 mM NaCl, 3 mM $MgCl_2$; 320 mM sucrose, pH 7.4) and centrifuged ($3500 \times g$, 5 min), twice, and the final pellet was retained.

**Quantitative proteomics**. Quadruplicate liver samples of the cytosol, mitochondria, and nucleus were homogenized, then lysed in lysis buffer (8 M urea, 50 mM Tris, pH = 8, 5 mM CaCl2, 20 mM NaCl, 1 EDTA-free Roche protease inhibitor tablet, and 1 Roche PhosSTOP phosphatase inhibitor tablet) with a probe sonicator for three pulses at 60 W, 20 kHz for 15 s, each followed by a 30 s pause for cooling at 4 °C. Crude lysates were then centrifuged at $14,000 \times g$ for 5 min, after which the supernatant was collected and protein concentrations were measured by Pierce BCA Protein Assay (Pierce, Rockford, IL, USA) according to the manufacture's protocol. Lysate containing 400 µg proteins was reduced in 5 mM dithiothreitol (DTT) at room temperature for 1 h, followed by alkylation in 15 mM iodoacetamide (IAA) for 30 min in the dark. Alkylation was quenched by adding DTT to 5 mM. The resulting solution was then diluted with Tris buffer (pH = 8) to 0.9 M urea, and proteins were digested with trypsin (Promega, Madison, WI) at 1:50 enzyme to protein ratio at 37 °C for 18 h. Digestion was quenched by adding trifluoroacetic acid (TFA) to a final concentration of 0.3% and desalted with C18 SepPak cartridges (Waters, Milford, MA, USA). Peptides were dried under vacuum and reconstituted in 0.5 M TEAB before labeling. For each genotype, samples were assigned to two batches of 4-plex dimethylated leucine (DiLeu) tags each in biological duplicate (two samples and two littermate controls per batch). In all, 4 mg of each DiLeu tags were suspended in anhydrous DMF and combined with 4-(4,6-dimethoxy-1,3,5-triazin-2-yl)-4-methyl-morpholinium tetrafluoroborate (DMTMM) and N-methylmorpholine (NMM) at 0.6 × molar ratios to tags. The mixture was vortexed at room temperature for 1 h. Following centrifugation, the supernatant was immediately mixed with 400 µg tryptic peptides from one condition. Peptides were labeled at a 10:1 label to peptide mass ratio and vortexed at room temperature for 2 h. The reaction was quenched by adding 5% NH₂OH to the final concentration of 0.25%, after which samples were dried under vacuum. Each batch of labeled peptides was combined, respectively, as 4-plex mixtures. The mixtures were purified by strong cation exchange liquid chromatography (SCX LC) with a PolySULFOETHYL A column (200 mm × 2.1 mm, 5 µm, 300 Å, PolyLC, Columbia, MD). Elutes containing labeled peptides were collected by a FC-4 fraction collector (Rainin Dynamax) and dried under vacuum. The cleaned samples were then fractionated with a Kinetex C18 column (5 µm, 100 Å, Phenomenex, Torrance, CA, USA), and a binary mobile phase at pH = 10 (mobile phase A is 10 mM aqueous ammonium formate and mobile phase B is 10 mM ammonium formate in 90% ACN). Gradient was set as following: 0–3 min 1% B phase; B phase linearly increased to 35% from 3 to 50 min, then quickly increased to 60, 70, and 100% in 4 min, 4 min, and 2 min, respectively, after which the column was washed by 100% B phase for 15 min. Eluents in the linear gradient were collected by 2-min intervals and combined into ten fractions. Each fraction was dried under vacuum. Peptides in each fraction were reconstituted in 0.1% formic acid (FA) and subjected to reversed phase LC-MS/MS analysis with an Orbitrap Fusion Lumos Tribrid mass spectrometer (ThermoFisher Scientific, San Jose, CA, USA) interfaced with a Dionex Ultimate 3000 UPLC system (ThermoFisher Scientific, San Jose, CA, USA). Peptides were loaded onto a 75-µm inner diameter microcapillary column custom-packed with 15 cm of Bridged Ethylene Hybrid C18 particles (1.7 µm, 130 Å, Waters). Labeled peptide were separated with a 90 min gradient from 3 to 30% ACN with 0.1% FA, followed by 10 min to 75% ACN and then 10 min to 95% ACN. After that, the column was equilibrated at 3% ACN for 15 min to prepare for the next injection. The mass spectrometer was operated in a top 20 data-dependent acquisition mode. Survey scans of peptide precursors from $m/z$ 350 to 1500 were performed at a resolving power of 60 K and an AGC target of $2 \times 10^5$ with a maximum injection time of 100 ms. The top 20 intense precursor ions were selected and subjected to the HCD fragmentation at a normalized collision energy of 30% followed by tandem MS acquisition at a resolving power of 15 K and an AGC target of $5 \times 10^4$, with a maximum injection time of 100 ms and a lower mass

limit of $m/z$ 110. Precursors were subjected to a dynamic exclusion of 45 s with a 10 ppm mass tolerance. Raw files were processed with Proteome Discoverer 2.1 engine (ThermoFisher Scientific, San Jose, CA, USA) with Byonic search engine (Protein Metrics Inc, San Carlos, CA, USA). Spectra were searched against the Uniprot *Mus musculus* reviewed database with trypsin as the enzyme and maximum two missed cleavages. The parent mass error tolerance was set to be 50 ppm, and fragment mass tolerance was 0.02 Da. Fixed modifications included DiLeu labels on peptide N-termini and lysine residues (+145.12801 Da) and carbamidomethylation on cysteine residues (+57.02146 Da). Dynamic modifications included oxidation of methionine residues (+15.99492 Da). Identifications were filtered to 1% peptide and protein FDR. Quantitation was performed in Proteome Discoverer with a reporter ion integration tolerance of 20 ppm for the most confident centroid. Only the PSMs that contained all reporter ion channels were considered, and protein quantitative ratios were determined using a minimum of one unique quantified peptide. Reporter ion ratio values for protein groups were exported to Microsoft Excel, and all fractions were combined for downstream analysis (see statistics section for processing). Proteins that had $P < 0.05$ (Fisher's method) were filtered as significant changes. The mass spectrometry proteomics data have been deposited to the ProteomeXchange Consortium via the PRIDE partner repository with the data set identifier PXD013736.

**Stoichiometry of acetylation**. Analysis was conducted using 200 μg of protein from the liver mitochondrial and cytosolic subcellular fractions, which were denatured in urea buffer (8 M urea (deionized), 500 mM ammonium bicarbonate pH = 8.0, 5 mM DTT)[28]. Samples were incubated for 20 min at 60 °C at 1500 rpm using the Eppendorf ThermoMixer® C. Cysteines were alkylated with 50 mM iodoacetamide and incubated for 20 min Chemical acetylation using two rounds of ~20 μmol heavy isotopic $D_6$-acetic anhydride (Cambridge Isotope Laboratories). The pH of the samples was spot checked and raised back to ~8.5 after each chemical labeling step. Samples were diluted with 100 mM ammonium bicarbonate pH = 8.0 to 2 M urea and digested with 1:100 trypsin at 37 °C for 4 h. Samples were then diluted to 1 M urea prior to a second digestion by gluC (1:100). Chemically acetylated peptides were resuspended into ~2 mL of HPRP buffer A (100 mM Ammonium Formate pH = 10) and injected onto a preequilibrated Phenomenex Gemini® NX-C18 column (5 μm, 110 Å, 150 × 2.0 mm) with 2% Buffer B (10% Buffer A, 90% acetonitrile). Peptides were separated with a Shimadzu LC-20AT HPLC system using a 2%–40% Buffer B linear gradient over 60 min at 0.4 mL/min flow rate, collecting 24 fractions throughout the length of the gradient. Fractions were dried down using a speedvac and pooled by concatenation into six final fractions. The samples were analyzed using data-independent acquisition (DIA) analysis by a Thermo Q-Exactive Orbitrap coupled to a Dionex Ultimate 3000 RSLC nano UPLC with a Waters Atlantic reverse phase column (100 μm × 150 mm). For data-independent acquisition (DIA), the MS survey scan was performed in profile mode with a resolution of 70,000, AGC of 1e6, maximum fill time of 100 ms in the scan range between 400 and 1000 m/z. The survey scan was followed 30 DIA scans in profile mode with a resolution of 35,000, AGC 1e6, 20 m/z window, and NCE of 30. The source voltage was set at 2000 V and capillary temperature at 250 °C. To deconvolute and analyze the DIA spectra, a spectral library containing all light and heavy acetyl-lysine feature pairs was generated. Spectral library samples were processed identically to the experimental samples, except they were treated with $C^{12}$-acetic anhydride (Sigma) and analyzed using data-dependent acquisition (DDA) mass spectrometry analysis. Using the openly available MaxQuant (v1.6.1) software package, we performed a database search to find peptides present in the DDA samples analyzed. Carbamidomethylation (C) was set as a fixed modification, and oxidation (M) and acetyl (K) were set as variable modifications. Trypsin and GluC were set as the digestion enzymes, with the max number of missed cleavages set to five. DDA runs from both the mitochondrial and cytosolic fractions were run to make one combined library. Heavy acetyl fragment ion pairs were generated in silico, such that the spectral library would contain both the light (endogenous) acetylation peaks and the heavy (chemical) acetylation peaks. The experimental samples were processed using Spectronaut (v10) using the generated spectral library. The subcellular fraction experimental samples were processed separately. The data were processed using an in-house R script, which can be accessed through the GitHub link: https://doi.org/10.5281/zenodo.3238525, such that stoichiometry was calculated from the ratio of endogenous (light) fragment ion peak area over the total (endogenous and chemical) fragment ion peak area. We performed an isotopic envelope correction of the heavy-labeled peak to remove any contribution from naturally occurring isotopes from the light labeled peak. All fractions were combined for downstream analysis; proteins that were $P < 0.05$ compared with WT were filtered as significant changes (see Statistics section for processing). The raw data, processed data, spectral library, and the analysis logs describing the settings for the Spectronaut analyses have been deposited to the ProteomeXchange Consortium via the MassIVE partner repository with the data set identifier PXD014013.

**Quantitative post-acquisition data set analysis**. Pathway analysis was performed using the online tool WebGestalt (www.webgestalt.org), using *Mus musculus* as the organism, Over-Representation Analysis as the method, and the KEGG pathway database. Uniprot identifiers of proteins passing a *p*-value threshold of 0.05 were used as the input and genome_protein-coding as the reference gene set. Additional

parameters were 5 as the minimum and 500 as the maximum number of genes for a category, Benjamini–Hochberg for multiple test adjustment to an FDR of 0.05.

To determine subcellular localization, proteins were categorized according to their primary Uniprot annotation. Secretory pathway localization includes the endoplasmic reticulum, Golgi apparatus, proteasome, lysosome, extracellular, and secreted proteins. The nucleus includes chromosome and nucleus. Cytoplasm includes cytoskeleton, cytosol, ribosomes, and all other organelles.

The heatmap was generated in Perseus (v. 1.5.8.5)[29] using $Log_2$ transformed fold change from the control. Hierarchical clustering was performed using Euclidean as the distance metric and complete as the linkage criterion.

Cluster analysis was determined using KEGG pathways, which arose during pathway analysis. For the lipid metabolism cluster, all proteins which were found in the following KEGG pathways in either of the AT-1 models and in either the proteome or acetyl stoichiometry were included: fatty acid degradation, peroxisome, butanoate metabolism, proponoate metabolism, biosynthesis of unsaturated fatty acid, and fat digestion. For the mitochondria-related cluster, all proteins that were found in the following KEGG pathways were included: PPAR signaling pathway, oxidative phosphorylation, TCA cycle, Alzheimer's disease, Parkinson's disease, and Huntington's disease. These proteins were the input for a STRING analysis. Proteins with no interactions were hidden, with the minimum required interaction score set at high confidence (0.7). Active interaction sources included all sources except textmining.

**Metabolomics**. Hepatocytes were plated at 900,000 cells per 35 mm collagen-coated plate. All cells received a media change of pre-media (10% FBS, 1% penicillin/streptomycin/glutamine in the Dulbecco's modified Eagle medium (DMEM)) 2 h before $^{13}C$ media was added. After the 2 h pre-incubation, cells were either immediately snap-frozen in liquid nitrogen or given $^{13}C$ media (10% FBS, 1 mM sodium pyruvate, 25 mM $^{13}C5$-labeled glutamine (99%, Cambridge Isotope Laboratories) in glutamine-free DMEM) for 30 min and then snap-frozen with liquid nitrogen. Plates of cultured hepatocytes and media were kept frozen at −80 °C until time of extraction. Media (500 μL) was lyophilized by vacuum centrifugation. Metabolites were extracted directly from plated hepatocytes or from lyophilized media on ice with 4 °C 7:2:1 HPLC-grade methanol:water:chloroform. After addition of extraction solvent, cells were immediately scraped from the plate using a cell scraper; media samples were vortexed and probe sonicated for 10 s. Resulting extracts were transferred to microcentrifuge tubes and incubated at 4 °C for 5 min, then centrifuged at $10000 \times g$ for 10 min at 4 °C to pellet precipitated protein. Extracts were aliquoted into glass autosampler vials and dried by vacuum centrifugation (150 μL for gas chromatography analysis). For GC-MS analysis, dried extract was derivatized for 90 min with 20 mg/mL methoxyamine hydrochloride in pyridine at 20 °C (25 μL) and then with MSTFA for 30 min at 37 °C (25 uL). Samples were analyzed by GC-Orbitrap; 1 μL of sample, split 1:10, was injected onto a TraceGOLD TG-5SilMS GC column (cat. no. 26096–1420, Thermo Scientific). Temperature was held at 50 °C for 1 min, then ramped to 320 °C at a rate of 11 °C/min, then held at 320 °C for 4.40 min. Molecules were analyzed with positive electron-impact (EI)-Orbitrap full scan of 50–650 m/z range. For data analysis, selected m/z and retention times were used to quantify metabolites and their isotopic distributes; peak areas were quantified using Thermo's Tracefinder application. We corrected for naturally occurring $^{13}C$ isotopes using tool written in MATLAB[30] and performed batch-correction using the ComBat function in R[31,32]. Metabolomics raw MS files are available on the public repository MassIVE with the accession code ID MSV000083885 (ftp://massive.ucsd.edu/MSV000083885). Additional information can be found in Supplementary Table 3 and Supplementary Data 3.

**Statistics**. Data analysis was performed in Graphpad Prism v 7.02 (GraphPad Software, Inc) and R v3.5.1. Unless otherwise specified, data are expressed as mean ± standard deviation. For the proteomics, fold changes were computed within each DiLeu batch experiment, an F-test was used to test for equivalent variance among groups, and a Student's *t* test was performed assuming equal or unequal variance according to the results of the F-test. A final fold change was calculated by averaging the two experiments together, and the *p*-values of the two separate DiLeu experiments were combined using Fisher's method as implemented in the R package *metap* (R version 3.5.1). For the stoichiometry of acetylation, a one-way ANOVA test was performed. Comparison of the proteins and acetyl proteoforms distribution was performed using a Kolmogorov–Smirnov test. For all other analyses, comparison of the means was performed using a Student's *t* test. The following statistical significance was used: *$P < 0.05$; **$P < 0.005$; #$P < 0.0005$.

**Reporting summary**. Further information on research design is available in the Nature Research Reporting Summary linked to this article.

## Data availability
The mass spectrometry proteomics data that support the findings of this study have been deposited in the ProteomeXchange Consortium via the PRIDE partner repository with the accession code PXD013736. The acetyl-proteomics data that support the findings of this study have been deposited to the ProteomeXchange Consortium via the MassIVE partner repository with the accession code PXD014013. The metabolomics data that

support the findings of this study have been deposited in the MassIVE repository with the accession code MSV000083885 (ftp://massive.ucsd.edu/MSV000083885).The R script that was used to process the acetyl-proteomics data have been deposited on Github with the identifier (search terms: AT1 Acetylation Stoich) (https://doi.org/10.5281/zenodo.3238525). The authors declare that all other data supporting the findings of this study are available within the paper and its supplementary tables.

## Code availability

The in house R script that was used to process the acetyl-proteomics data have been deposited on Github with the identifier (search terms: AT1 Acetylation Stoich) (https://doi.org/10.5281/zenodo.3238525). The README file found on Github describes how the input data for the scripts can be accessed through ProteomeXchange accession code PXD014013.

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

## Acknowledgements

We thank Dr. John Svaren for critical reading of an early version of this paper. We thank Karl Miller for technical help. This work was supported by the NIH (NS094154 and AG053937 to L.P.; AG057408 to L.P. and R.M.A.; GM65386 to J.M.D.; R01DK071801, R01AG052324, and P41GM108538 to L.L.; P41 GM108538 to J.J.C.); a core grant to the Waisman Center from NICHD-U54 HD090256. I.A.D. was supported by T32 AG000213 and T32 GM007507. SIM imaging was performed at the Biochemistry Optical Core of the University of Wisconsin-Madison (Madison, WI). The Orbitrap instruments were purchased through the support of an NIH shared instrument grant (NIH-NCRR S10RR029531) and Office of the Vice Chancellor for Research and Graduate Education at the University of Wisconsin-Madison. This work was also supported using resources and facilities of the William S. Middleton Memorial Veterans Hospital (Madison, WI, USA).

## Author contributions

I.A.D., A.J.L., Y.P., Q.Y., K.A.O., Y.C., E.A.A. and P.R.H. performed the experiments and analyzed the data. T.W.R. analyzed the data. M.S.B., L.L., J.M.D., J.J.C., R.M.A. and L.P. provided critical advice for the experiments. L.P. designed the overall study. L.P. and I.A.D. wrote the paper with input from all authors.

## Additional information

**Competing interests:** J.M.D. is a co-founder of Galilei BioScience Inc, devoted to the development of small-molecule effectors for SIRT6. Remaining authors have no conflict of interests to disclose.

