## [Peer Review File · Nature Communications]

Reviewers' comments:

Reviewer #1 (Remarks to the Author):

NCOMMS-19- 06663-T

This manuscript reports on an exciting, landmark discovery that demonstrates a global regulation of hepatocyte function by the trafficking of cytosolic acetyl-CoA. Hepatic acetyl-CoA transport was modulated in vivo in 2 mouse models that either had reduced AT-1-mediated transport of acetyl-CoA from the cytosol to the ER lumen, or enhanced transport of acetyl-CoA from the cytosol to the ER lumen. Liver lipid metabolism adapted to the enforced restriction or enhancement of acetyl-CoA transport from the cytosol and was associated with alterations in protein expression and protein acetylation. The quantitative proteome/acetylome analyses included a very nice filtering strategy that implicated a subset of cellular processes that were responsive to cytosolic acetyl-CoA flux. The alterations of mitochondrial 3-D morphology in the AT-1 sTg mouse liver were remarkable. This report provides a platform for further, more detailed analyses of the molecular connections that mediate the subcellular crosstalk in which acetyl-CoA plays a major role. This is original work that will guide further investigation within the emerging field of CoA metabolism.

It is a surprise that manipulation of the AT-1 transport function would alter the cytosolic proteome and acetylome so dramatically, particularly since the cytosolic acetyl-CoA steady-state levels that were measured (Fig. 1, panels f and m) did not differ substantially in the 2 mouse models. The exception was the AT-1 sTg liver on the control diet (panel m) where there was a modest reduction in acetyl-CoA levels compared to WT. So a question arises about the acetyl-CoA crosstalk between subcellular organelles, particularly in light of the mitochondrial acetylome responses (Fig. 4, panel c).

Do the acetyl-CoA levels in unfractionated cells or in isolated mitochondria differ between the mouse models or the diets?

Are there acetyl-CoA transporters located in membranes of other organelles?

The higher fatty acid oxidation capacity in the AT-1 sTg hepatocytes (Fig. 1, panel o) was difficult to reconcile with the 'flux' data (Fig. 4, panel g) where there were no discernible differences between the levels of TCA cycle intermediates in WT and sTg liver. Meanwhile the higher fatty acid oxidation capacity was in general agreement with the resistance to lipid droplet accumulation on a high carb diet. If not developed further, the 'flux' data could be deleted from the study.

An interpretation of the 'flux' data for both the AT-1^{+/+} and AT-1sTg hepatocytes would be helpful. The conversion of alpha-ketoglutarate to succinate requires CoA and so is CoA limiting in the AT-1^{+/+} hepatocytes?

The 'flux' was measured at a single time point – 30 min., but shouldn't flux be measured at several time points to indicate rate of conversion? Perhaps 30 min was an in appropriate time point and the ¹³C-labelled pools may have already equilibrated, keeping in mind that several of the TCA cycle reactions are reversible. The readout for the measurement of fatty acid oxidation capacity was oxygen consumption, so one would expect a greater rate of TCA cycle activity in the AT-1 sTg hepatocytes, unless.....

Why was flux measured with ¹³C-glutamine/glutamate? Would ¹³C-fatty acid be a more appropriate probe to evaluate a correspondence with greater fatty acid oxidation capacity and enrichment of lipid metabolism enzymes?

The alterations in hepatic lipid metabolism, e.g., lipid droplet formation, are quite dramatic in the AT-1^{-/+} mice. Both free fatty acids and triglycerides would accumulate in the droplets and these lipids are significantly elevated (Fig. 1, panel f). The elevation of free fatty acids suggests a depressed capacity for acyl-CoA formation.

Do the fatty acyl-CoA levels differ between WT and AT-1^{-/+} liver? Acyl-carnitine levels?

The serum cholesterol was markedly reduced in the AT-1 sTg mice on both the control and the high

carb diets (Fig. 1, panel n). These data indicate reduced cholesterol synthesis from acetyl-CoA which would occur at the ER/Golgi membrane followed by lipoprotein assembly in the ER lumen and secretion. Does AT-1 overexpression divert acetyl-CoA toward protein acetylation in the ER lumen at the expense of cholesterol synthesis? This would be a point for possible discussion or future investigation.

Reviewer #2 (Remarks to the Author):

Dieterich and colleagues have examined the effects of AT-1 deficiency or overexpression on metabolic phenotypes in mice. The studies provide a good deal of proteomic analysis of the tissue of these models and the findings are consistent with role of this transporter in affecting protein acetylation. The mice also seem to have interesting metabolic phenotypes, but this is tough to interpret and incompletely explored. Thus, while this is an interesting observational paper, the cause and effect relationships are incompletely explored.

Although there's a good deal of data suggesting that altered protein acetylation is the mechanistic explanation for the observed metabolic phenotype, the cause and effect relationship between these observations is not demonstrated. It is plausible as well that the effects on acetyl-CoA in the cytosol are also having direct effects on metabolism by affecting flux of acetyl-CoA into other pathways such as de novo lipogenesis and/or TCA cycle.

There are a number of points that are not internally or externally consistent. Previous work on mitochondrial protein acetylation has pretty much always suggested that increased acetylation is associated with reduced fat oxidation, while the opposite is suggested here. The ¹³C-glutamine flux experiments would suggest that TCA activity is increased in the deficient cells, which isn't consistent with the overall metabolic phenotype or the overexpression data.

A number of the metabolic analyses are incomplete or lacking key details in their description. I have noted some specific examples but the methods should be improved with more details in general:

1. The Seahorse analysis lacks key details in the concentrations of the nutrients in the media as well as the concentrations of the inhibitors used. There is no mention in the results of the data generated with the pyruvate transport or other inhibitors.
2. The 'fluxomic' data is very minimally done and overly simplistic. The ¹³C flux data only indicate two TCA cycle metabolites were analyzed; one of which is generated by enzymatic steps outside of the TCA cycle. Is there increased flux or is this because there is a metabolic bottleneck further in the TCA cycle (e.g. citrate synthase). Additional metabolites should be reported. It is not clear what alpha-KG is being measured. Is this M+5 or has this been through multiple rounds of the cycle? In supplementary table 3, how could there be an M+7 form of this when there are only 5 carbons in the molecule?
3. Measurement of only one gene to explain the changes in proteomic is very superficial.

The severe phenotype of the overexpressing mice (previously reported) make the data generated with these mice tough to interpret. It would have greatly improved this work to have a tissue-specific overexpression system.

Minor:

Although the results state that acetyl-Coa is increased in the deficient liver, but the values are not significantly different.

Reviewer #3 (Remarks to the Author):

The manuscript by Dieterich et al. titled "Acetyl-CoA flux regulates the proteome and acetyl-proteome to maintain intracellular metabolic crosstalk" is an interesting and comprehensive assessment of multi-omics changes specifically investigating the role of acetyl CoA transporter. In addition to proteomics changes the authors assess acetylation changes, lipid changes, and acetylation stoichiometry, and some metabolic flux measurements. Overall the measurements are technically good and interesting, however raw data should be provided as well as more detailed processed data, so not just global plots are shown but results may also be presented in more distilled tables. I recommend revisions addressing the below comments:

Specific Comments:

- The authors present stoichiometry data (occupancy) – the method should indicate some more specifics on the data processing for stoichiometry. For the calculations of stoichiometry were precursor ions on the MS1 level used to determine the light to heavy ratios or did the authors use MS2 DIA fragment ions (differentiating fragment ions) that contain the Lys-acetyl group to obtain relevant ratios of light/heavy. Please provide more details. Also the MS section describing DIA should contain more acquisition details.
- Supplemental table 2 shows log ratio of stoichiometry changes between AT1 vs WT and AT1 mutant vs WT, however the authors should also list the stoichiometry values itself – so readers can see which sites do have the highest stoichiometry in %, which is different from just seeing a delta change. For Supplemental table 2 peptide sequences and other MS details should also be provided, m/z etc. For the PTM analysis it is important to list the acetylated peptides with sequence and modification site indicated in the supplement (particularly as a double proteolytic system is apparently used trypsin/GluC), so the field can see in more details what analysis was performed to yield these results.
- For the study presented here some of the stoichiometry data should be confirmed with an independent assay, for selected sites, that may show a high stoichiometry, or that are relevant biologically the authors could purchase synthetic peptides (modified and unmodified, heavy) and confirm with an SRM or PRM assay to validate the stoichiometry measurements.
- Also that way some biological relevance could maybe be demonstrated for some of the results and interesting PTM sites. Currently, no real biological follow up is performed based on the measurements (specifically the stoichiometry measurements).
- Overall the data presentation contains no MS specific details. Details are currently not provided for the identifications nor for the quantification (ID: peptide sequences, m/z, mass accuracies, for protein level what peptides were identified; for PTM level what are the sequences of the modified peptides; also quantification details should be provided such as Spectronaut outputs or Proteome discoverer outputs). If the latter tables are too large they can also be provided as data upload to repositories. Supplemental table 1 is also very sparse and does not contain how many peptides were used for quantification etc.
- A summarizing table showing the subsets of protein changes and PTM site stoichiometry changes shown in Figure 4b and c would be helpful as this seems to be a key subset of proteins and PTM sites that they authors present as most interesting. This may be currently distributed in provided tables as part of the overall comprehensive results – it would be interesting to split out this subset.
- Similarly for the flux data – processed results should be presented as supplemental tables.
- In general all raw data should be uploaded to public repositories (such as Pride, ProteomeXchange, Massive...)
- MS/MS spectra of acetylated peptides that are featured as relevant or significant should be provided as part of a spectral library or MSMS viewer, or uploaded to MS viewer software, i.e. Panorama... .

POINT-BY-POINT RESPONSE

We thank the Editor and the Reviewers for the positive evaluation of our work. We have now addressed all the comments. A detailed point-by-point response can be found below. In brief: (i) we have deposited all of the raw data sets in publicly available repositories; (ii) we have updated the Supplementary Tables to show the subsets of protein changes (Supplementary Table 1), the post-translation modification (PTM) site stoichiometry changes (Supplementary Table 2), and the list of processed metabolites (Supplementary Table 3); (iii) we have expanded Fig. 4g to include more metabolites; and (iv) we have expanded the Methods section of the manuscript. As of now, this revised manuscript includes 4 figures and 5 Supplementary Tables. Changes within the manuscript are highlighted.

Reviewer 1

...since the cytosolic acetyl-CoA steady-state levels that were measured (Fig. 1, panels f and m) did not differ substantially in the 2 mouse models. The exception was the AT-1 sTg liver on the control diet (panel m) where there was a modest reduction in acetyl-CoA levels compared to WT.

Response: Taken out of context, the steady-state levels of cytosolic acetyl-CoA only offer a limited view of the metabolic changes because the liver can adapt. Specifically: (a) AT-1^{S113R/+} mice adapt to the increased availability of cytosolic acetyl-CoA by diverting excess acetyl-CoA into fatty acids, triglycerides and very large lipid droplets (this adaptive response is quite dramatic; see Fig. 1). Therefore, it was not unexpected that the steady-state levels of cytosolic acetyl-CoA in AT-1^{S113R/+} mice were only about 60% higher than WT. (b) AT-1 sTg mice, in contrast, adapt to the depletion of cytosolic acetyl-CoA by “importing” more acetyl-CoA through glycolysis and β -oxidation (see also our Aging Cell 2018 paper). This adaptive response -in both models- partially compensates for the changes in acetyl-CoA availability caused by the genetic modification.

The higher fatty acid oxidation capacity in the AT-1 sTg hepatocytes (Fig. 1, panel o) was difficult to reconcile with the ‘flux’ data (Fig. 4, panel g) where there were no discernible differences between the levels of TCA cycle intermediates in WT and sTg liver...If not developed further, the ‘flux’ data could be deleted from the study.

Response: The ¹³C isotope tracing data are not meant to explain the liver phenotype or the fatty acid oxidation capacity. They are meant to show that the TCA cycle can adapt to changes in AT-1 activity (and acetyl-CoA flux from the cytosol to the ER lumen). We performed this study with both ¹³Cglucose and ¹³Cglutamine at “time 0” (to detect absolute levels) and “time 30” (to confirm the time 0 data with a labeled precursor). Even in this case, when taken out of context, the isotope tracing studies do not say much. The fundamental conclusion of Fig. 4 is that AT-1, an ER membrane transporter, can force “mitochondria reprogramming” by altering the cytosolic levels of acetyl-CoA. Indeed, we reported changes at multiple levels: mitochondria proteome, mitochondria acetylome, mitochondria morphology, and TCA cycle engagement. When taken together, these results are very novel and have the potential to be paradigm-shifting (further discussed later).

An interpretation of the ‘flux’ data for both the AT-1+/- and AT-1sTg hepatocytes would be helpful. The conversion of alpha-ketoglutarate to succinate requires CoA and so is CoA limiting in the AT-1+/- hepatocytes?

Response: AT-1 is an antiporter. The transfer of acetyl-CoA (from the cytosol to the ER lumen) is coupled to the transfer of free CoA (from the ER lumen to the cytosol). Therefore, we could certainly envision a reduced availability of CoA in the cytosol (and we do have some indirect evidence). However, this is difficult to document and, certainly, beyond the scope of this manuscript.

The 'flux' was measured at a single time point – 30 min., but shouldn't flux be measured at several time points to indicate rate of conversion? Perhaps 30 min was an in appropriate time point and the 13C-labelled pools may have already equilibrated, keeping in mind that several of the TCA cycle reactions are reversible. The readout for the measurement of fatty acid oxidation capacity was oxygen consumption, so one would expect a greater rate of TCA cycle activity in the AT-1 sTg hepatocytes, unless.....

Response: As mentioned above, we measured TCA engagement with both 13Cglucose and 13Cglutamine at “time 0” (to detect absolute levels) and “time 30” (to detect 13C metabolites). Time 30 (with 13C labeling) was necessary to confirm the time 0 data. Therefore, we are not trying to measure rate of conversion of individual metabolites. The rate of conversion of individual TCA metabolites is well established and it is unlikely to be informative for our work. The Reviewer is correct in mentioning that (i) the 30 min is an appropriate time point because the 13C pools are already equilibrated and (ii) several TCA cycle reactions are reversible. Furthermore, we must also consider that (iii) the product of each reaction is also the substrate of the next and (iv) some of these metabolites can exit the TCA cycle and the mitochondria through different membrane transporters and equilibrate. That is why we refrained from over-interpreting the 13C isotope tracing data. [REDACTED]

Why was flux measured with 13C-glutamine/glutamate? Would 13C-fatty acid be a more appropriate probe to evaluate a correspondence with greater fatty acid oxidation capacity and enrichment of lipid metabolism enzymes?

Response: As mentioned above, we used 13C isotope tracing to determine engagement of the TCA cycle and not fatty acid oxidation capacity. Therefore, 13C-fatty acids would not be an appropriate method.

The alterations in hepatic lipid metabolism, e.g., lipid droplet formation, are quite dramatic in the AT-1-/+ mice. Both free fatty acids and triglycerides would accumulate in the droplets and these lipids are significantly elevated (Fig. 1, panel f). The elevation of free fatty acids suggests a depressed capacity for acyl-CoA formation.

Response: We did not look at rate of acyl-CoA formation because beyond the scope of our work. As the Reviewer points out, the lipid droplet formation is very dramatic and consistent with the increased levels of fatty acids and triglycerides. In essence, these results clearly show the impact that AT-1 activity has for lipid metabolism. This is one of the main points of our manuscript and is evident from Fig. 1.

Does AT-1 overexpression divert acetyl-CoA toward protein acetylation in the ER lumen at the expense of cholesterol synthesis? This would be a point for possible discussion or future investigation.

Response: We cannot comment on this question – at least for now. It could certainly be a topic of future investigation. [REDACTED]

Reviewer 2

Although there's a good deal of data suggesting that altered protein acetylation is the mechanistic explanation for the observed metabolic phenotype, the cause and effect relationship between these observations is not demonstrated. It is plausible as well that the effects on acetyl-CoA in the cytosol are also having direct effects on metabolism by affecting flux of acetyl-CoA into other pathways such as de novo lipogenesis and/or TCA cycle.

Response: We totally agree with the Reviewer. Indeed, we report changes in the proteome, acetylproteome, TCA engagement and lipid metabolism. Nowhere in our manuscript we have claimed that altered protein acetylation is the single mechanistic explanation for the phenotype.

There are a number of points that are not internally or externally consistent. Previous work on mitochondrial protein acetylation has pretty much always suggested that increased acetylation is associated with reduced fat oxidation, while the opposite is suggested here.

Response: The opposite relation between mitochondria acetylation and lipid oxidation is also true here. AT-1 sTg mice have reduced mitochondria acetylation (Fig. 3h, yellow dots), increased fatty oxidation (Fig. 1o) and are very resistant to diet-induced steatosis. However, this dual relationship between acetylation of mitochondria proteins and fatty oxidation cannot be taken as a general rule (in the sense that it does not apply to every experimental model/paradigm).

The 13C-glutamine flux experiments would suggest that TCA activity is increased in the deficient cells, which isn't consistent with the overall metabolic phenotype or the overexpression data.

Response: The main point of Fig. 4 is to demonstrate that changes in rate of acetyl-CoA transport across the ER membrane can force mitochondria adaptation. Indeed, we report changes at multiple levels: mitochondria proteome, mitochondria acetylproteome, mitochondria morphology, and TCA cycle engagement. When taken together, these results are very novel.

[REDACTED]

[REDACTED]

1. *The Seahorse analysis lacks key details in the concentrations of the nutrients in the media as well as the concentrations of the inhibitors used. There is no mention in the results of the data generated with the pyruvate transport or other inhibitors.*

Response: We have expanded the description of the Seahorse analysis in the Methods section. The use of alternate pathway inhibitors (UK5099 and BPTES) was used to generate fatty acid oxidation capacity. The equation used to determine this is now included in the Methods as well. We thank the Reviewer for noticing this.

2. *The 'fluxomic' data is very minimally done and overly simplistic. The ¹³C flux data only indicate two TCA cycle metabolites were analyzed; one of which is generated by enzymatic steps outside of the TCA cycle. Is there increased flux or is this because there is a metabolic bottleneck further in the TCA cycle (e.g. citrate synthase). Additional metabolites should be reported. It is not clear what alpha-KG is being measured. Is this M+5 or has this been through multiple rounds of the cycle? In supplementary table 3, how could there be an M+7 form of this when there are only 5 carbons in the molecule?*

Response: We have included more metabolites (see current Fig 4g of our revised manuscript). We have also updated a new version of Supplementary table 3. Finally, we have provided detailed explanation of the isotope tracing interpretation above. Regarding, alpha-KG question, please bear in mind that the formula of the ion that we quantify contains more than 5 carbons (C₁₁H₂₂NO₅Si₂); this is because we use derivitization with MSTFA and quantify the GC-MS fragment ion containing the 5 carbons of Alpha-KG + 2 TMS groups. To account for natural isotopic abundance (~1% ¹³C), we initially quantify 2+ more isotopes than expected and then perform natural isotope abundance correction.

3. *Measurement of only one gene to explain the changes in proteomic is very superficial.*

Response: We did not look at PGC1 α mRNA to explain the proteomic changes and we never attempted to make that connection. PGC1 α is a major transcriptional regulator of mitochondria biogenesis and the "natural" target every time there is an expansion of the mitochondria network. The levels of PGC1 α (Fig. 4f) reflect the changes reported in Fig. 4d-e. We should also point out that we also looked at new "emerging" mitochondria regulators such as CLUH but we did not observe any significant change.

Minor:

Although the results state that acetyl-Coa is increased in the deficient liver, but the values are not significantly different.

Response: As mentioned above (under Reviewer 1), the steady-state levels of cytosolic acetyl-CoA only offer a limited view of the metabolic changes because the liver is adapting. Specifically: (a) AT-1^{S113R/+} mice adapt to the increased availability of cytosolic acetyl-CoA by diverting excess acetyl-CoA into fatty acids, triglycerides and very large lipid droplets (this adaptive response is quite dramatic; see Fig. 1). Therefore, it was not unexpected that the steady-state levels of cytosolic acetyl-CoA in AT-1^{S113R/+} mice were only about 60% higher than WT. (b) AT-1 sTg mice, in contrast, adapt to the depletion of cytosolic acetyl-CoA by "importing" more acetyl-CoA through glycolysis and β -oxidation (see also our Aging Cell 2018 paper). This adaptive response -in both models- partially compensates for the changes in acetyl-CoA availability caused by the genetic modification.

Reviewer 3

The authors present stoichiometry data (occupancy) – the method should indicate some more specifics on the data processing for stoichiometry. For the calculations of stoichiometry were precursor ions on the MS1 level used to determine the light to heavy ratios or did the authors use MS2 DIA fragment ions (differentiating fragment ions) that contain the Lys-acetyl group to obtain relevant ratios of light/heavy. Please provide more details. Also the MS section describing DIA should contain more acquisition details.

Response: The methods section has been updated to include more of the sample preparation and DIA acquisition details. In regard to the specific question about quantification, we use the MS2 DIA fragment ions, from which we identify both the heavy and light labeled Lys-acetyl group. The full details of this method are being published in a separate journal but can currently be accessed through the pre-print database, BioRxiv (<https://www.biorxiv.org/content/10.1101/472530v1.article-info>). In addition to including more of the data collection details, we have also attached the R-scripts used to quantify stoichiometry from the Spectronaut output files. These can be found deposited on GitHub using the link: <http://doi.org/10.5281/zenodo.3238525>. All necessary scripts, input files, and output files are included. We have also added the peptide sequence and other relevant details to our stoichiometry Supplementary table 2.

Supplemental table 2 shows log ratio of stoichiometry changes between AT1 vs WT and AT1 mutant vs WT, however the authors should also list the stoichiometry values itself – so readers can see which sites do have the highest stoichiometry in %, which is different from just seeing a delta change. For Supplemental table 2 peptide sequences and other MS details should also be provided, m/z etc. For the PTM analysis it is important to list the acetylated peptides with sequence and modification site indicated in the supplement (particularly as a double proteolytic system is apparently used trypsin/GluC), so the field can see in more details what analysis was performed to yield these results.

Response: To address this concern, we made two changes. First: Supplementary Table 2 now has peptide sequence and overall acetylation stoichiometry. Second: because we use a heavy/light labeling scheme along with precursor and fragment ions of acetylated sites, we have made available the Spectronaut output files, which can be found in both the ProteomeXchange link as well as in the R data analysis folder. We have also included 2 files containing the m/z ratios for our precursor ions (PrecMz_supplement.csv) and fragment ions (FrgMz_supplement.csv) found in the spectral library used to deconvolute the DIA analyses. These files can be found in the ProteomeXchange link (see below). These demonstrate that for a single acetylation site there are multiple groups of the precursor and fragment ions used to calculate overall stoichiometry.

For the study presented here some of the stoichiometry data should be confirmed with an independent assay, for selected sites, that may show a high stoichiometry, or that are relevant biologically the authors could purchase synthetic peptides (modified and unmodified, heavy) and confirm with an SRM or PRM assay to validate the stoichiometry measurements.

Response: We have performed extensive validation of our improved stoichiometry method. The full details of this method are being published in a separate journal but can currently be accessed through the pre-print database, BioRxiv

(<https://www.biorxiv.org/content/10.1101/472530v1.article-info>). We refer the reviewer to the data in Figure 2 of this manuscript.

Also that way some biological relevance could maybe be demonstrated for some of the results and interesting PTM sites. Currently, no real biological follow up is performed based on the measurements (specifically the stoichiometry measurements).

Response: One of the main conclusions of this paper is that changes in acetyl-CoA transport across the ER membrane cause increased predisposition (AT-1^{S113R/+} mice) or resistance (AT-1 sTg mice) to aberrant lipid accumulation within the cytosol. This was associated with changes at the level of the proteome, acetylome or both that occurred within specific metabolic pathways that impinge on the above increased predisposition or resistance. Therefore, the changes in proteome and/or acetyl-proteome provide the mechanistic connection to the actual phenotype (hence, the general biological relevance of the changes reported – if this is what the Reviewer is referring to with “biological relevance”).

Nε-lysine acetylation has been proposed to affect many aspects of protein metabolism, including, conformation, stability, sorting, activity, etc. of targeted proteins. Therefore, a comprehensive biochemical validation of each identified modified residue is a major undertaking and certainly beyond the scope of this paper (if this is what the Reviewer is referring to with “biological relevance”). It is also important to point out that it would be very difficult to test the biochemical output of a pathway where a certain enzyme was affected at the protein level (either increased or reduced levels) while another enzyme was affected at the acetyl-proteome level (either increased or reduced acetylation). This would be further complicated by the fact that the acetylation of different Lys residues might be doing different things. Therefore, even if we decided to mutate a specific Lys residue of a specific enzyme for which a clear *in vitro* biochemical assay exists, we would not be able to come up with a generalized interpretation of the acetyl-proteome data.

Overall the data presentation contains no MS specific details. Details are currently not provided for the identifications nor for the quantification (ID: peptide sequences, m/z, mass accuracies, for protein level what peptides were identified; for PTM level what are the sequences of the modified peptides; also quantification details should be provided such as Spectronaut outputs or Proteome discoverer outputs). If the latter tables are too large they can also be provided as data upload to repositories. Supplemental table 1 is also very sparse and does not contain how many peptides were used for quantification etc.

Response: Please see response to comments above. For the acetyl-proteome, the raw data and the spectronaut output files can be found in the ProteomeXchange files. We have updated Supplementary table 1 show the number of unique peptides found for each protein. For this study we were interested in protein level abundance, instead of peptide level abundance. However we have provided the peptide identification and quantifications as a part of the ProteomeXchange repository (ID: PXD013736).

A summarizing table showing the subsets of protein changes and PTM site stoichiometry changes shown in Figure 4b and c would be helpful as this seems to be a key subset of proteins and PTM sites that they authors present as most interesting. This may be currently distributed in provided tables as part of the overall comprehensive results – it would be interesting to split out this subset.

Response: An appropriate modified Supplementary Table (see Supplementary Table 5) has been included to this revised version of the manuscript.

Similarly for the flux data – processed results should be presented as supplementary tables.

Response: This was done as well (see new Supplementary Table 3).

In general all raw data should be uploaded to public repositories (such as Pride, ProteomeXchange, Massive...)

Response: We have updated the submission to include all raw data, details regarding processing steps and detailed output files. For the acetyl-proteomics, all raw data, Spectronaut outputs, spectral libraries, and analysis details have been uploaded to ProteomeXchange via MassIVE.

Acetyl-proteomics ID: PDX014013 (accessible once published)
Acetyl-proteomics Massive ID: MSV000083855
Username: MSV000083855_reviewer
Password: AT1acstoich19

For the proteomics, all raw data files and peptide sequence data have been uploaded to ProteomeXchange.

Proteomics ID: PDX013736
Username: reviewer94184@ebi.ac.uk
Password: JEDucov0

For the metabolomics, raw files are available on the public repository MassIVE

Metabolomics ID: MSV000083885

MS/MS spectra of acetylated peptides that are featured as relevant or significant should be provided as part of a spectral library or MSMS viewer, or uploaded to MS viewer software, i.e. Panorama... .

Response: We have now provided all the raw Spectronaut output files containing the information the reviewer requested. However, if the reviewer still wants to view the spectral data via a MS viewer, we will be happy to accommodate the reviewer.

ADDITIONAL GENERAL COMMENTS:

R script code submission: The R script has been made publicly available at the following GitHub link: <http://doi.org/10.5281/zenodo.3238525>

- The code and software submission checklist (nr-software-policy.pdf) has been submitted as a related manuscript file.

m/z details for acetyl-proteomics: These files have been submitted as related manuscript files [precursor ions (PrecMz_supplement.csv) and fragment ions (FrgMz_supplement.csv)]

- These file can also be found using the following information:
Acetyl-proteomics ID: PDX014013 (accessible once published)
Acetyl-proteomics Massive ID: MSV000083855

Username: MSV000083855_reviewer
Password: AT1acstoich19

Data Availability: We have added a data availability statement after the methods section with all necessary data citations.

Code Availability: We have added a code availability statement in the methods section with the doi identifier.

REVIEWERS' COMMENTS:

Reviewer #1 (Remarks to the Author):

The authors have done a nice job in responding to the comments of Reviewer 1. The responses clarified many points that are directly relevant to the interpretation of the data in this paper. The text of the revised manuscript is adapted to help the reader understand the relationships between the proteomic and biochemical results. Some of the questions posed by Reviewer 1 were not addressed directly but Reviewer 1 agrees that these questions are beyond the scope of this paper. In fact, some of the comments were written to stimulate future research directions.

The genetic mouse models demonstrate a causative relationship between cytosolic acetyl-CoA regulation and the adaptive proteomic responses that lead to development, or lack of development, of a hepatic steatosis phenotype. The proteomic adaptations to altered AT-1 transport activity are the main focus and absolutely novel and unexpected. The additional data destined for future publication that are generously included in the response indicate the authors' vision for development of the concepts that are presented here for the first time.

There is a huge amount of data and a few recommendations for insertions in the text may help the cursory reader understand the gist of the conclusions:

Line 104

... proteins that were significantly altered...

insert

... expression levels of proteins that were significantly altered....

Line 113

... these translational changes, ...

suggested modification:

... these protein expression changes, ...

Line 124

... unique translational consequences ...

Suggested modification

... unique protein expression consequences ...

The protein expression levels change but the authors have not demonstrated that the changes are solely translational and could also involve protein degradation or turnover, and/or transcriptional changes.

Line 219

... Cpt1a to be modified at ...

Insert descriptors - How are these proteins modified? Does expression or acetylation increase or decrease?

Line 222

... acetylome level integrate the oxidation of fatty acids ...

Insert

... acetylome level integrate to promote the oxidation of fatty acids ...

Suzanne Jackowski, Ph.D.

Reviewer #2 (Remarks to the Author):

My concerns were addressed

Reviewer #3 (Remarks to the Author):

The authors took great care to add a lot of additional details and extra information which is good.

REVIEWERS' COMMENTS:

Reviewer #1 (Remarks to the Author):

Line 104 ... proteins that were significantly altered... insert ... expression levels of proteins that were significantly altered...

Response: Modified as requested.

Line 113 ... these translational changes, ... suggested modification: ... these protein expression changes, ...

Response: Modified as requested.

Line 124 ... unique translational consequences ... Suggested modification ... unique protein expression consequences ... The protein expression levels change but the authors have not demonstrated that the changes are solely translational and could also involve protein degradation or turnover, and/or transcriptional changes.

Response: Modified as requested.

Line 219 ... Cpt1a to be modified at ... Insert descriptors - How are these proteins modified? Does expression or acetylation increase or decrease?

Response: This information is incorporated in the Supplementary Information and in the Data Sets.

We refrained from discussing this too much in the manuscript because the integration of proteomic and acetylomic changes is quite complex (and largely unknown). Any attempt to interpret it in terms of “increased” or “reduced” would be very limiting (and -at this stage- not informative). For example, stating that the acetylation of K192 in AT-1 sTg has increased while the acetylation of K455 in Hadha has decreased does not translate in any immediate biochemical conclusion. What is really important (and this what we say in the Results section) is the fact that the increased fatty acid oxidation capacity of AT-1 sTg is reflected in changes of the carnitine-acylcarnitine translocase machinery and the hydroxyacyl-CoA dehydrogenase/3- ketoacyl-CoA thiolase/enoyl-CoA hydratase trifunctional enzyme, and that these changes occur at the level of the proteome and acetylome. The concept that a metabolic pathway might be regulated by integrating changes that affect the proteome and/or the acetylome is very novel and will certainly challenge the field. However, if the Reviewer or the Editors feel that we should discuss this further, we can try expanding the section.

Line 222 ... acetylome level integrate the oxidation of fatty acids ... Insert ... acetylome level integrate to promote the oxidation of fatty acids ...

Response: modified as requested